# The dynamic assembly of distinct RNA polymerase I complexes modulates rDNA transcription

Eva Torreira[1†], Jaime Alegrio Louro[1†], Irene Pazos[2,3], Noelia González-Polo[4], David Gil-Carton[5], Ana Garcia Duran[2,3], Sébastien Tosi[2,3], Oriol Gallego[2,3*], Olga Calvo[4*], Carlos Fernández-Tornero[1*]

[1]IPSBB Unit, Centro de Investigaciones Biológicas, Madrid, Spain; [2]Institute for Research in Biomedicine, Barcelona, Spain; [3]The Barcelona Institute of Science and Technology, Barcelona, Spain; [4]Instituto de Biología Funcional y Genómica, CSIC-Universidad de Salamanca, Salamanca, Spain; [5]Structural Biology Unit, Cooperative Center for Research in Biosciences CIC bioGUNE, Derio, Spain

**Abstract** Cell growth requires synthesis of ribosomal RNA by RNA polymerase I (Pol I). Binding of initiation factor Rrn3 activates Pol I, fostering recruitment to ribosomal DNA promoters. This fundamental process must be precisely regulated to satisfy cell needs at any time. We present in vivo evidence that, when growth is arrested by nutrient deprivation, cells induce rapid clearance of Pol I–Rrn3 complexes, followed by the assembly of inactive Pol I homodimers. This dual repressive mechanism reverts upon nutrient addition, thus restoring cell growth. Moreover, Pol I dimers also form after inhibition of either ribosome biogenesis or protein synthesis. Our mutational analysis, based on the electron cryomicroscopy structures of monomeric Pol I alone and in complex with Rrn3, underscores the central role of subunits A43 and A14 in the regulation of differential Pol I complexes assembly and subsequent promoter association.

*For correspondence: oriol.gallego@irbbarcelona.org (OG); ocalvo@usal.es (OC); cftornero@cib.csic.es (CF-T)

[†]These authors contributed equally to this work

Competing interests: The authors declare that no competing interests exist.

## Introduction

The nucleolus constitutes a cellular hub dedicated to ribosome biogenesis, which starts with the transcription of ribosomal RNA (rRNA) precursor genes by RNA polymerase I (Pol I). Ribosomes are completed by the action of Pol II, synthesizing messenger RNA, and Pol III, involved in 5S rRNA and transfer RNA production. The critical requirement for ribosomes in actively growing cells causes that Pol I retains up to 60% of the total transcriptional activity within the eukaryotic nucleus (*Warner, 1999*). However, under stress conditions, cells tune down ribosome biosynthesis by repressing Pol I transcription (*Mayer et al., 2004*). Accordingly, defects in the regulation of this process can lead to uncontrolled cell proliferation and have been associated with different types of cancer (*Bywater et al., 2012*).

Initiation of rRNA synthesis in yeast is a sequential process that involves four components: the upstream activating factor (UAF) complex, the TATA box-binding protein (TBP), the core factor (CF) heterotrimer and the Rrn3 protein (*Moss et al., 2007*). While the first three components recognize different regions in promoter DNA, Rrn3 binding to Pol I is a prerequisite for enzyme recruitment to promoter-bound initiation factors (*Keener et al., 1998*; *Yamamoto et al., 1996*). Except for UAF, an equivalent set of proteins plays similar roles in mammals (*Hannan et al., 1999*), where Rrn3 is also termed TIF-IA (*Bodem et al., 2000*; *Moorefield et al., 2000*). The recent electron cryomicroscopy (cryo-EM) structures of the Pol I–Rrn3 complex show how these two components interact

(*Engel et al., 2016*; *Pilsl et al., 2016*), confirming previous biochemical studies (*Blattner et al., 2011*; *Peyroche et al., 2000*).

Yeast Pol I is a 590 kDa enzyme composed of 14 subunits, whose atomic architecture was recently revealed (*Engel et al., 2013*; *Fernández-Tornero et al., 2013*). The two largest subunits, A190 and A135, forming the DNA-binding cleft and harbouring the active centre, are held together by the AC40/AC19 assembly heterodimer. Five rather globular subunits present in all nuclear RNA polymerases (Rpb5, Rpb6, Rpb8, Rpb10 and Rpb12) attach on the periphery of the complex. Subunit A12.2 contains a TFIIS-like C-terminal zinc ribbon located next to the active site, while the A49/A34.5 heterodimer attaches on the Pol I lobe through a TFIIF-like dimerization module. Finally, the A43/A14 heterodimer forms a stalk that emerges from the enzyme core.

In the Pol I crystal structure, the enzyme forms homodimers that exhibit an unexpectedly open cleft occupied by a DNA-mimicking loop, which is incompatible with transcription. The structure suggests that the interaction between the A43 C-terminal tail and the opposite monomer's clamp is important to maintain Pol I dimers. Interestingly, Pol I dimers in solution present the same structural arrangement, as shown by cryo-EM (*Pilsl et al., 2016*). While homodimerization has been proposed as a potential regulatory mechanism of Pol I activity, no evidence is available to date showing Pol I dimers to exist within the cell.

Here, we have investigated the molecular mechanisms underlying Pol I activation using a holistic approach. Live-cell imaging shows that nutrient depletion causes rapid clearance of Pol I–Rrn3 complexes, followed by the formation of Pol I homodimers, while nutrient addition reverts both repressive conditions following a quasi-symmetric pattern. Additionally, we report the cryo-EM structures of monomeric Pol I alone and in complex with Rrn3 at 4.9 and 7.7 Å resolution, respectively, representing the two steps in Pol I activation. Finally, we designed stalk mutants affecting Pol I dimerization and/or Rrn3 binding, and used them to explore the influence of these complexes on rDNA association.

## Results

### Live-cell imaging of Pol I complexes

To investigate the formation and disruption of Pol I transcription complexes in vivo, we used PICT (Protein interactions from Imaging Complexes after Translocation), a fluorescence microscopy technique to analyse protein interactions in living cells (*Gallego et al., 2013*; *Picco et al., 2017*). This technique uses cellular anchoring platforms tagged with both RFP and FK506-binding protein (anchor-RFP-FKBP) to recruit proteins tagged with FKBP-binding domain (bait-FRB). FKBP and FRB strongly interact in the presence of rapamycin, which induces translocation of the bait-FRB to the anchor. If a GFP-tagged protein (prey-GFP) interacts with the bait, it will co-translocate to the anchor upon rapamycin addition, leading to increased co-localization of RFP and GFP signals. Previously-engineered anchors generate a large number of anchors at the plasma membrane, which only allows the detection of abundant cytosolic complexes. We designed a new anchor by tagging Tub4, a component of the spindle pole body, with RFP and FKBP (Tub4-RFP-FKBP). The resulting cells harbour only one or two anchors, thus increasing the PICT sensitivity by up to 200-fold (*Figure 1—figure supplement 1A*; *Video 1*). In addition, Tub4 is exposed to both the cytosol and the nucleus, which allows detection of complexes on both sides of the nuclear envelope (*Figure 1—figure supplement 1B*). The levels of recruitment can be quantified as the co-localization between prey-GFP and

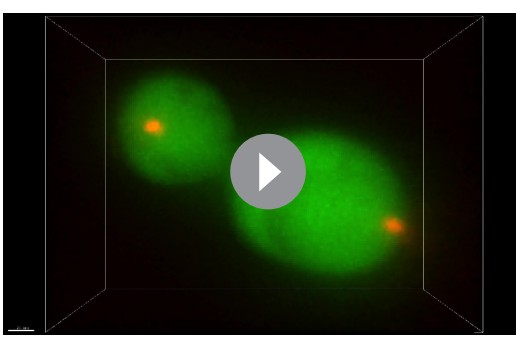

**Video 1.** Engineered anchoring platform associated with the spindle pole body. Yeast cell expressing GFP and Tub4-RFP-FKBP imaged in z-stacks of 250 nm incremental steps. Imaris software was used to obtain the 3D reconstruction. A maximum of two anchoring platforms could be observed in each cell.

anchor-RFP-FKBP (see 'Materials and methods'; *Figure 1—figure supplement 2*).

Since Pol I transcription is down-regulated by rapamycin, all subsequent experiments were performed in rapamycin-insensitive strains carrying the *tor1-1* mutation (*Helliwell et al., 1994*), so that the addition of this compound has no effect on Pol I association to rDNA promoters (*Figure 1—figure supplement 3A*).

## Pol I dimerization is induced by nutrient deprivation and depends on A43 C-terminus

To investigate whether Pol I is able to form homodimers in vivo, we constructed a diploid strain where the Pol I subunit A190 was labelled in an allele-specific manner, with one allele tagged to GFP (A190-GFP) and the second to FRB (A190-FRB). The presence of these tags did not alter the doubling times of the cells (101.6 ± 11.6, 103.0 ± 6.2 and 101.2 ± 12.3 min for the parental, A190-GFP and A190-FRB strains, respectively). When rapamycin is added, GFP-labelled Pol I will only co-translocate to the anchor-RFP-FKBP if it interacts with FRB-tagged Pol I (*Figure 1A*). Cells incubated in rich medium present normal growth rate and the vast majority of Pol I accumulates in a sub-nuclear structure likely corresponding to the nucleolus (*Figure 1—figure supplement 3B*). Upon rapamycin addition, no recruitment of A190-GFP could be detected at the anchors (*Figure 1B*; *Figure 1—figure supplement 3C*). In contrast, when cells were incubated in a medium lacking carbon and nitrogen, hereafter starving medium, their growth was arrested and A190-GFP translocated to the nuclear side of the anchors (*Figure 1B*; *Figure 1—figure supplement 1B*). Interestingly, the total levels of A190 as well as the distribution of A190-GFP in the anchor vicinity prior to rapamycin addition are equivalent in both media (*Figure 1—figure supplement 3D–E*). In accordance, the levels of bait recruitment are equivalent in growing and starved cells (*Figure 1—figure supplement 3F*). In addition, we performed co-immunoprecipitation experiments after crosslinking, using a diploid strain where one A190 allele was tagged with TAP (A190-TAP) and the second with MYC (A190-MYC). The former was used for pull-down with IgG resin while the latter was employed for western-blot analysis with anti-MYC antibody. Whole cell extracts (WCE) showed that A190-MYC immunoprecipitation is similar for cells incubated in rich (R) or starving (ST) medium (*Figure 1—figure supplement 4A*, lanes 1–4). Centrifugation of whole cell extracts allowed separation of a soluble fraction (SF) from a chromatin-associated insoluble fraction (Chr F), which were examined independently. Analysis of the soluble fraction showed that Pol I homodimers are only detected in starved cells (lanes 5 and 6). As expected, the chromatin insoluble fraction of growing cells presented high levels of A190-MYC (lane 7), likely corresponding to rDNA-associated Pol I molecules, while tiny amounts of A190-MYC were detected for starved cells (lane 8). DNase I treatment of the latter indicates that this is due to minor levels of Pol I that remains associated with DNA after two hours of starvation (lane 9). The absence of histone H3 in the soluble fraction indicates that there is no contamination from the chromatin insoluble fraction (*Figure 1—figure supplement 4B*).

The crystal structure of inactive Pol I identified the A43 C-terminal tail, encompassing residues 260–326, as a key element to form Pol I homodimers (*Figure 1A*, inset). Therefore, we studied Pol I dimerization upon partial deletion of this structural element (A43ΔCt, Δ307–326). In this mutant, Pol I homodimerization is impaired (*Figure 1B*), confirming the observation derived from structural data.

To evaluate whether RNA polymerase dimerization is a more general regulatory mechanism, we applied PICT analysis to the other nuclear RNA polymerases. In the case of Pol III, no oligomerization was observed on the anchor in either growing or starving medium, indicating that Pol III does not dimerize in these conditions (*Figure 1—figure supplement 5*). Similarly, we were unable to detect Pol II oligomerization. However, accurate quantification was difficult in this case due to strong Pol II-GFP signal in the surroundings of the nuclear envelope where the anchor is located, which could mask recruitment of Pol II-GFP.

## Defects downstream of rRNA synthesis trigger Pol I homodimerization

To rule out the possibility that Pol I homodimerization could involve the synthesis of new proteins, we followed the formation of Pol I dimers in the presence of cycloheximide, which targets ribosomes and inhibits protein synthesis. Starved cells exposed to this compound did not show any defect in Pol I homodimerization. Instead, we detect a 40% increase in the homodimers levels with respect to untreated cells. This indicates that no additional protein synthesis is required to induce this cellular

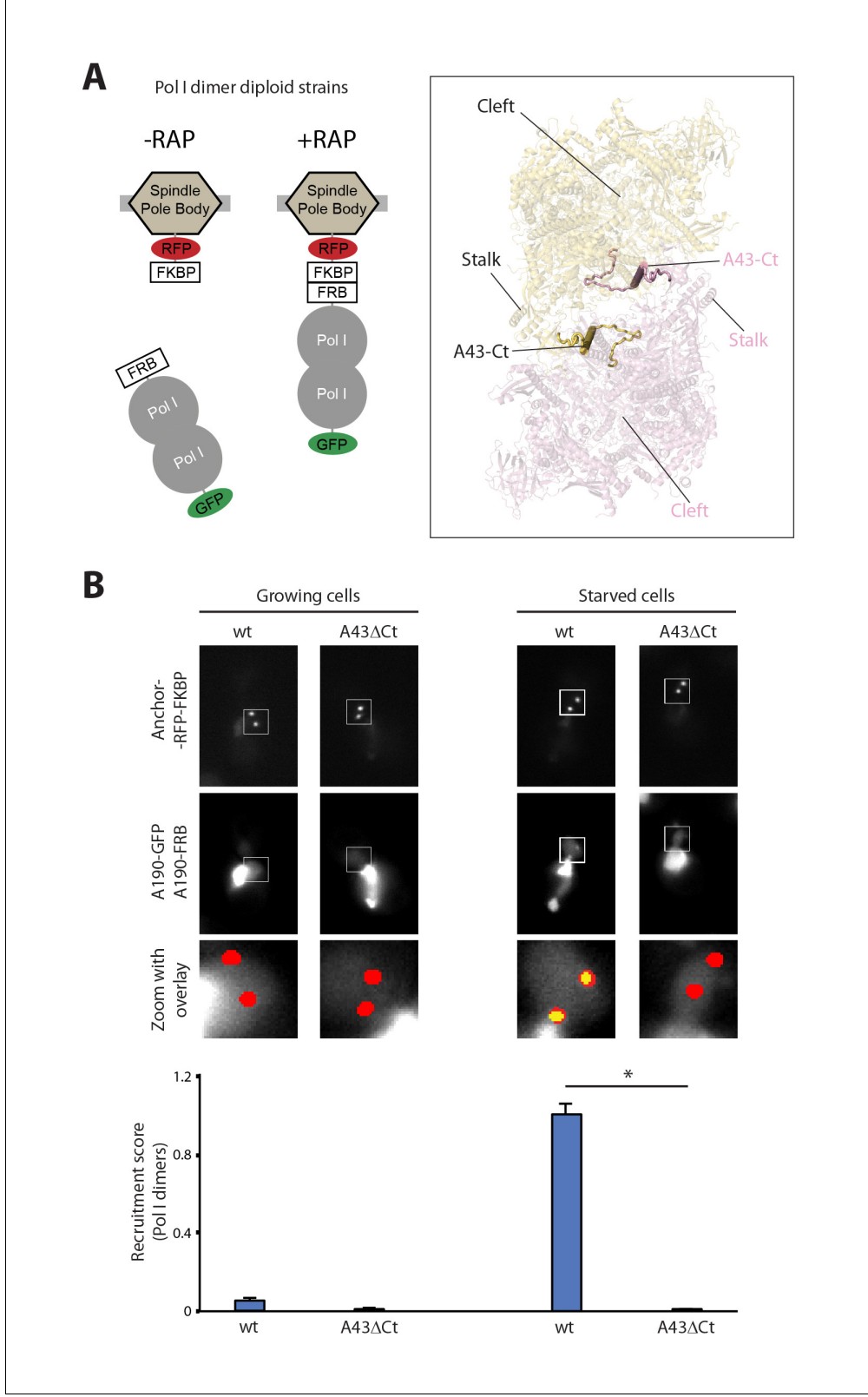

**Figure 1.** Live-cell imaging of Pol I homodimerization. (**A**) Scheme of the diploid strains designed to study Pol I homodimers in vivo. The crystal structure of inactive Pol I homodimers, critically maintained by the A43 C-terminal tail, is shown on the right with monomers in yellow and pink. (**B**) Representative PICT images of the RFP-tagged anchor (upper row), GFP-tagged A190 (middle row) and a zoom of a 2.6 × 2.6 μm square around the anchoring

*Figure 1 continued on next page*

*Figure 1 continued*

platforms (bottom row). Below, quantification of the A190-GFP recruitment score, normalized to the measurement of the wild-type strain in starving medium (Mean ± SD, p-value * < 0.01 t-test).

The following figure supplements are available for figure 1:

**Figure supplement 1.** A more sensitive PICT assay to detect cytosolic and nuclear complexes.
**Figure supplement 2.** Methodology for PICT quantification.
**Figure supplement 3.** Additional control experiments.
**Figure supplement 4.** Co-immunoprecipitation on Pol I dimerization.
**Figure supplement 5.** Analysis of Pol II and Pol III complexes.

event and that ribosome inhibition could reinforce Pol I dimerization (*Figure 2A*). Interestingly, inhibition of protein synthesis in cells grown in non-starving medium was sufficient to induce Pol I dimerization (*Figure 2B*). When cycloheximide was replaced by diazaborine, an inhibitor of rRNA maturation during 60S formation (*Loibl et al., 2014*), Pol I dimers were also assembled (*Figure 2B*). These results show that Pol I homodimerization is induced by inhibition of either protein synthesis or rRNA maturation, two processes that are downstream of rRNA synthesis.

## Nutrient depletion induces Pol I–Rrn3 clearance

The Pol I–Rrn3 complex represents the activated form of the enzyme (*Milkereit and Tschochner, 1998*). We used PICT to evaluate whether cells also modulate the levels of this complex according to nutrient availability. We thus constructed a haploid strain expressing anchor-RFP-FKBP, Rrn3 labelled with GFP (Rrn3-GFP) and A190-FRB (*Figure 3A*; *Figure 3—figure supplement 1A*). Again, the presence of these tags did not alter the doubling time of the cells (101.6 ± 11.6 and 100.7 ± 15.03 min, for the parental and tagged strains). Despite Pol I–Rrn3 complexes were detected in cells incubated in both rich and starving media, their levels were reduced to about 40% upon nutrient deprivation (*Figure 3B*). Importantly, while starvation does not induce significant changes in the distribution of Rrn3-GFP in the vicinity of the anchor (*Figure 3—figure supplement 1B*), the cellular levels of Rrn3-GFP drop significantly in starved cells (*Figure 1—figure supplement 3D*). The reduction in Rrn3 and Pol I–Rrn3 correlates with an almost complete deprivation of both Pol I and Rrn3 from rDNA promoters, as observed by ChIP experiments (*Figure 3C–D*, wt-35S). In the case of Pol I, a similar drop in association is also observed inside the rDNA gene, where Rrn3 is absent even in wild-type cells (*Figure 3D*, wt-18S and wt-25S).

To further investigate the effect of starvation on Pol I transcription, we used our A43 C-terminal truncation abolishing Pol I dimerization. For that, we constructed A43ΔCt haploid cells that, while expressing similar levels of Rrn3 and A190 to wild-type cells, present reduced growth (*Figure 3—figure supplement 1C*). Structural data show that the A43 C-terminal tail is not involved in Rrn3 binding (see below; *Engel et al., 2016*; *Pilsl et al., 2016*). Growing A43ΔCt cells exhibit levels of Pol I–Rrn3 complexes equivalent to the wild-type (*Figure 3B*), whereas Pol I promoter association is slightly increased (*Figure 3D*). A greater increase in Pol I occupancy is observed inside the rDNA gene under growing conditions (*Figure 3D*). More interestingly, in starved A43ΔCt cells, the levels of Pol I–Rrn3 complexes are about 2-fold higher than those detected in wild-type cells (*Figure 3B*). Furthermore, in these conditions, association of Pol I along the rDNA gene and of Rrn3 at the promoter increase about 6-fold with respect to the wild-type (*Figure 3D*).

We also used a strain, termed CARA for Constitutive Association of Rrn3 and A43, expressing a Pol I–Rrn3 chimera that cannot form Pol I homodimers (*Laferté et al., 2006*). ChIP experiments show that the Pol I–Rrn3 chimera is normally associated to rDNA in growing cells (*Figure 3—figure supplement 2A*). However, under starving conditions, Pol I–Rrn3 association along rDNA is 6 to 7-fold higher than in wild-type cells, a similar behaviour to A43ΔCt cells. Moreover, recovery of CARA from nutrient-depleted medium is faster than for wild-type cells (*Figure 3—figure supplement 2B*).

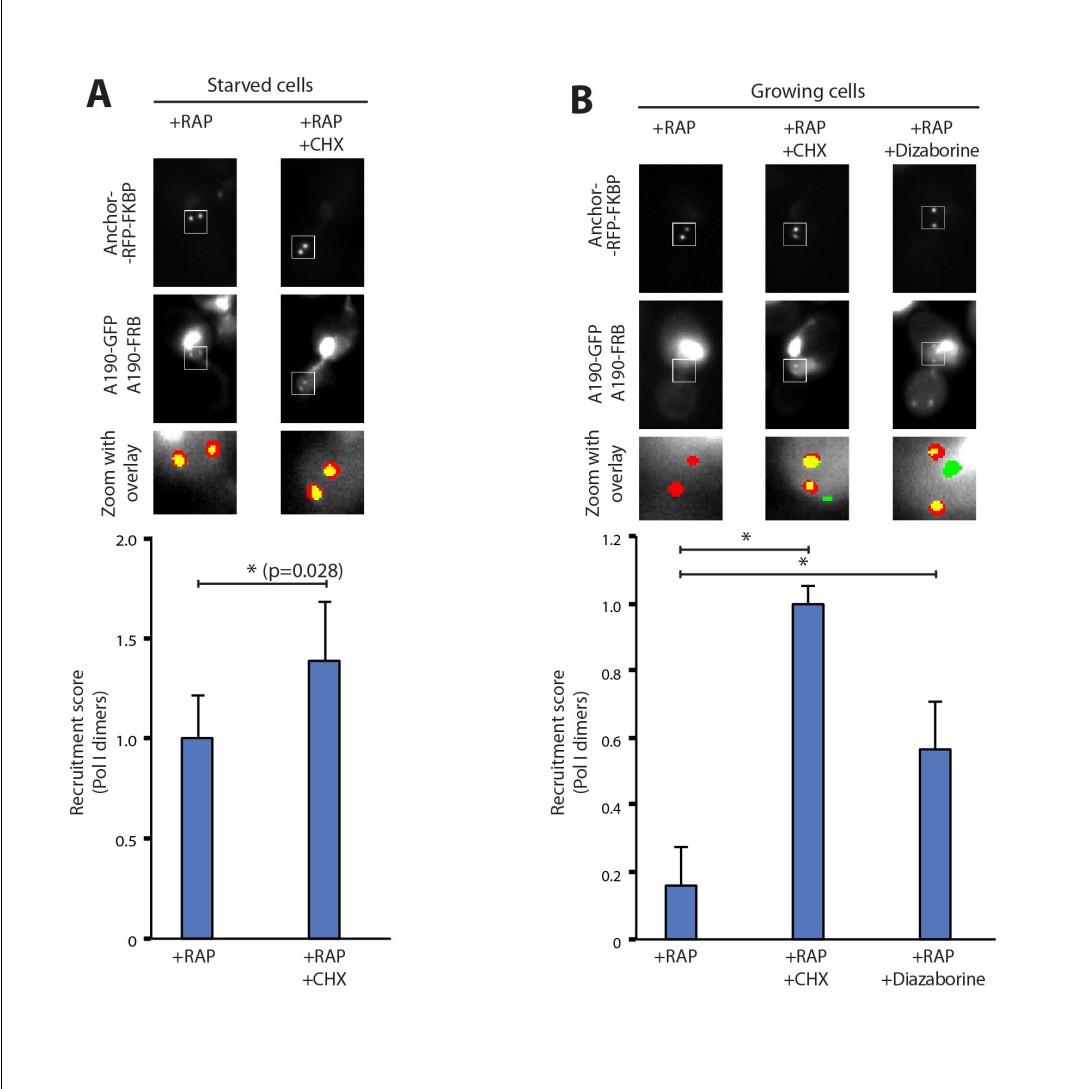

**Figure 2.** Pol I homodimerization upon ribosome perturbation. Representative PICT images of the RFP-tagged anchor (upper row), GFP-tagged A190 (middle row) and a zoom of a 2.3 × 2.3 µm square around anchoring platforms (bottom row). Below, quantification of the A190-GFP recruitment score, normalized to the measurement of untreated cells (Mean ± SD, p-value * < 0.01 t-test). (**A**) Effect of cycloheximide (CHX; 0.2 µg/ml) in starved cells. (**B**) Effect of cycloheximide (0.2 µg/ml) and diazaborine (10 µg/ml) in growing cells.

Overall, our results indicate that Pol I homodimerization is important for complete Pol I–Rrn3 clearance and transcription inactivation upon starvation.

## Dynamics in the assembly of Pol I complexes in response to nutrient availability

We then quantified the temporal progression in the assembly and disassembly of Pol I inactive dimers and Pol I–Rrn3 active complexes in response to nutrient availability. When growing cells are transferred to starving medium, detected levels of Pol I–Rrn3 are rapidly depleted by about 30%, in a process that follows an exponential decay (*Figure 4A*, left). Accordingly, the rate of complex clearance is maximal within the first 15 min, while Pol I homodimers remain undetectable at this stage. In a second stage, disassembly of Pol I–Rrn3 complexes slows down, reaching an additional 10% decrease within the next 20 min (i.e. 40% total reduction when compared to growing cells). During this period, the observed homodimerization follows a sigmoid-like tendency, reaching its fastest assembly rate. After 35 min of starvation, cells enter a third stage where both Pol I–Rrn3 clearance

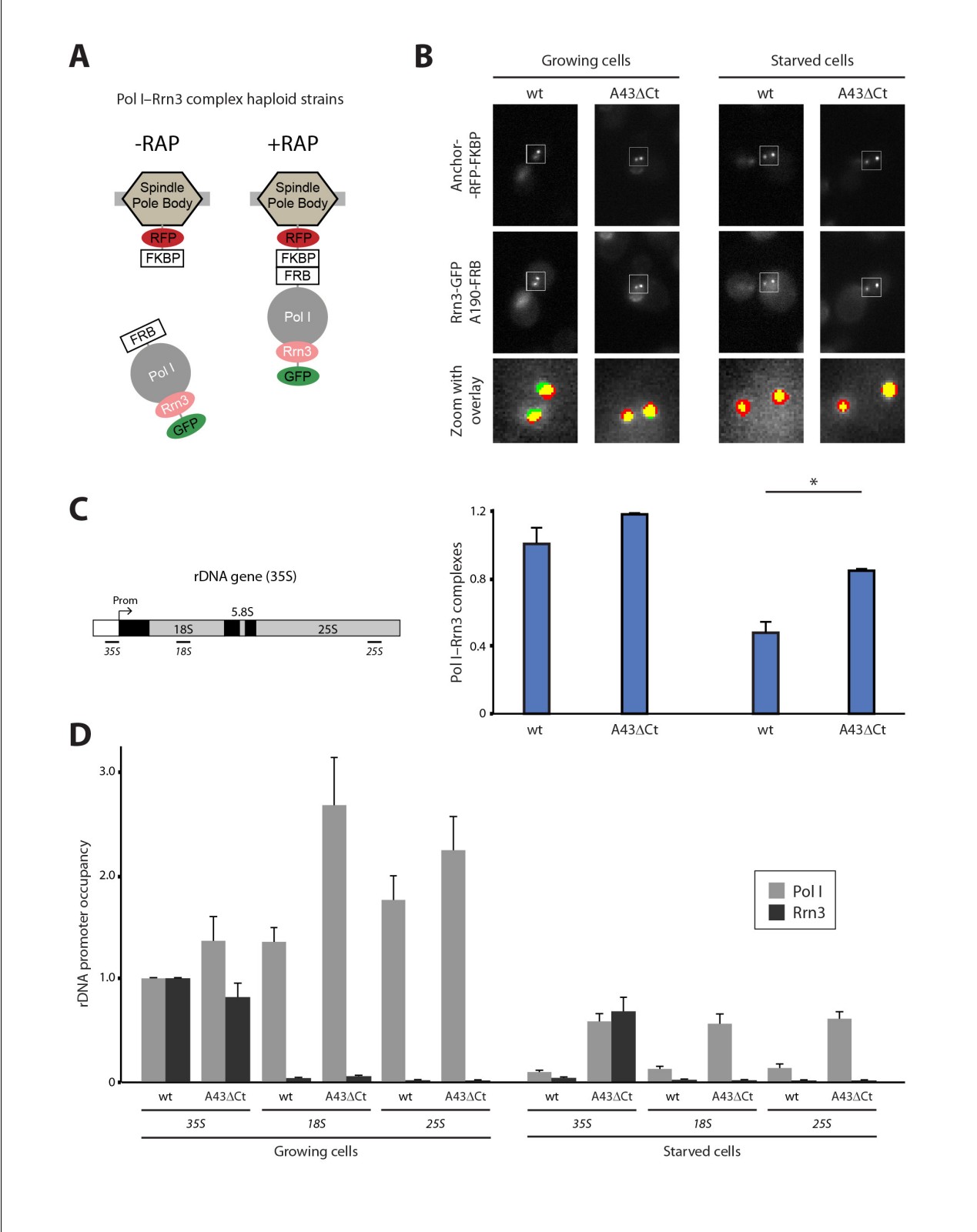

**Figure 3.** Live-cell imaging of Pol I–Rrn3 complexes. (A) Scheme of the haploid strains designed to study Pol I–Rrn3 in vivo. (B) Representative PICT images of the RFP-tagged anchor (upper row), GFP-tagged Rrn3 (middle row) and a zoom of a 2.3 × 2.3 μm square around anchoring platforms (bottom row). Below, quantification of the Rrn3-GFP recruitment score, normalized to the measurement in rich medium (Mean ± SD, p-value * < 0.01

*Figure 3 continued on next page*

*Figure 3 continued*

t-test). (C) Schematic representation of the 35S rDNA. Below, approximate location of the primer pairs used for ChIP experiments in the following panel. (D) ChIP experiments showing the relative occupancy of A190 and Rrn3 on the rDNA gene in different culture media (Mean ± SD).

The following figure supplements are available for figure 3:

**Figure supplement 1.** Additional control experiments.

**Figure supplement 2.** Analysis of the CARA strain.

and Pol I homodimerization rates remain slow but constant. Here, detected Pol I–Rrn3 complexes drop by an additional 20% (i.e. 60% total reduction) while the levels of Pol I homodimers double in amount. A symmetric pattern is observed when starved cells are transferred to nutrient rich medium (*Figure 4A*, right). Cells initially respond through rapid assembly of Pol I–Rrn3 during the first 15 min, where detected levels exponentially increase up to 50% of the maximum value in rich medium. Remarkably, changes in the levels of Pol I homodimers are fast as well, leading to 30% reduction in this initial stage. In a second stage, detected Pol I–Rrn3 complexes remain constant while the clearance rate of Pol I homodimers is maximal, resulting in undetectable levels of Pol I homodimers after 35 min from nutrient addition. From this point, the assembly of Pol I–Rrn3 complexes is slowly restored to the level observed in normal growing conditions.

ChIP experiments performed at an equivalent regime show that the promoter association of both A190 and Rrn3 drops by two thirds within the first 10 min from starvation but requires about 2 hr to reach completion (*Figure 4B*, left). Whereas this exponential tendency is comparable to that of Pol I–Rrn3 clearance, a significant amount of this complex can still be detected in spite of undetectable levels of A190 and Rrn3 on rDNA promoters, as mentioned above. In contrast, restoration of rDNA promoter association follows a linear pattern that is accomplished in about 2 hr (*Figure 4B*, right). Western-blot analysis at the same time-points shows that the overall levels of A190 remain constant while those of Rrn3 correlate with detected amounts of Pol I–Rrn3 (*Figure 4—figure supplement 1*), suggesting that Rrn3 levels influence the number of Pol I–Rrn3 complexes, as previously proposed (*Philippi et al., 2010*). Overall, these results indicate that cells respond to nutrient availability by differentially adjusting the levels of Pol I homodimers and Pol I–Rrn3.

## Cryo-EM structures of monomeric Pol I alone and in complex with Rrn3

To unveil the molecular details of Pol I activation, we aimed to characterize this process structurally. Negatively-stained 2D averages of the Pol I–Rrn3 complex showed a portion of additional density next to the stalk (*Figure 5—figure supplement 1A*). Antibody labelling confirms that this density corresponds to Rrn3, which binds Pol I with its N-terminus facing the stalk (*Figure 5—figure supplement 1B–C*). We then studied the Pol I activation process using cryo-EM (*Figure 5—figure supplement 2A–B*). Unsupervised 3D classification into six classes identified two interesting groups with about 90,000 and 32,000 particles (*Figure 5—figure supplement 2C*). Refinement of the first subset, which corresponds to monomeric Pol I, yielded a 5.6 Å resolution map (*Figure 5—figure supplement 2C*, right). Refinement of the second group, containing an extra piece of density next to the Pol I stalk that corresponds to Rrn3, produced a 7.7 Å resolution map (*Figure 5—figure supplement 2C*, left). Finally, the addition of both particle sets followed by 3D refinement yielded a map that is virtually identical to that of monomeric Pol I but reaches a resolution of 4.9 Å (*Figure 5—figure supplement 2C*, central), thus allowing the building of a quasi-atomic model (*Figure 5A*; *Figure 5—figure supplement 3*).

Our monomeric Pol I structure provides a detailed picture of the structural transition from homodimers to monomers, a critical step in Pol I activation (*Figure 5A–B*). The Pol I stalk appears almost completely disordered, in spite of weak density at the region directly contacting the enzyme core, i.e. the tip domain in subunit A43 (*Figure 5C*). This indicates that the stalk is highly dynamic, which may be important in the Pol I activation process. Additionally, the A12.2 C-terminal Zn-ribbon, the central part of the bridge helix and the DNA-mimicking loop are not visible in our structure, suggesting that these regions are also flexible (*Figure 5C*). When compared with the crystal structure of

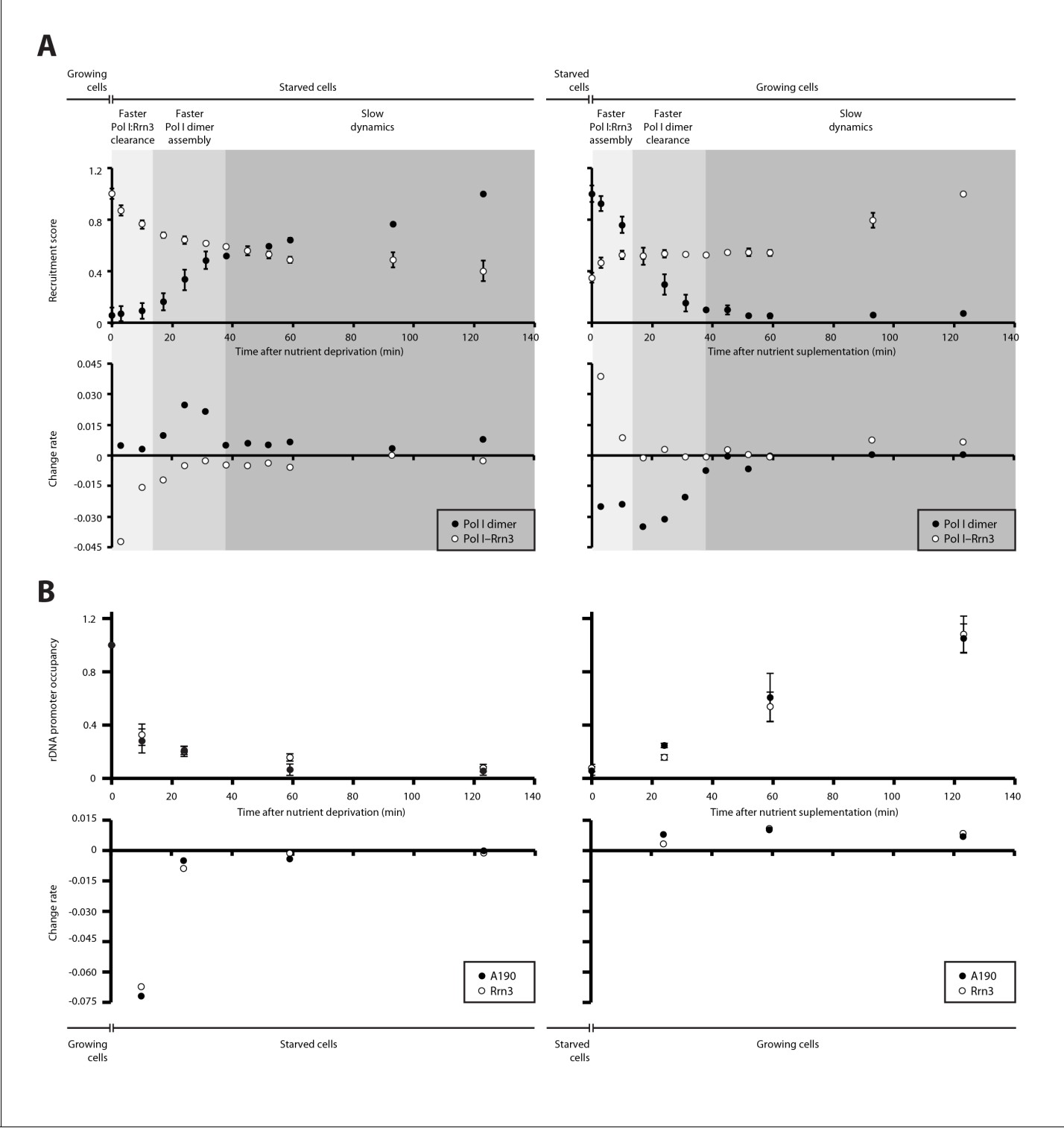

**Figure 4.** Dynamics in the cellular levels of Pol I complexes in response to nutrient availability. (**A**) Upper plots show the relative levels of Pol I–Rrn3 complexes and Pol I homodimers as detected by PICT. Measurements were done at the indicated time points after growing cells were switched to starving medium (left) or cells starved for 2 hr were switched to nutrient-rich medium (right). Values were normalized to the highest measurement of the corresponding complex. Bottom plots show the rate of assembly (positive values) and disassembly (negative values) of each complex between two consecutive measurements. Grey shadows indicate the different stages observed in response to nutrient availability: faster adjustment of Pol I–Rrn3 levels (lighter grey), faster adjustment of Pol I homodimer levels (middle grey) and slow consolidation of the levels of each Pol I complex (dark grey). (**B**) Upper plots show the relative A190 and Rrn3 occupancy at the rDNA promoter for the indicated conditions and time points, as measured by ChIP.

*Figure 4 continued on next page*

*Figure 4 continued*

Values were normalized to the value obtained for cells growing in rich medium (Mean ± SD). Bottom plots show the rate of assembly and disassembly, calculated as in panel **A**.

The following figure supplement is available for figure 4:

**Figure supplement 1.** A190 and Rrn3 levels in response to nutrient availability.

dimeric Pol I (*Engel et al., 2013*; *Fernández-Tornero et al., 2013*), which is essentially identical to dimeric Pol I in solution (*Pilsl et al., 2016*), our monomeric Pol I structure presents a rearranged cleft entrance where the clamp coiled-coil and the protrusion approach by about 4 Å (*Figure 5B*). The resulting cleft entrance is 38 Å in width, which leaves enough room for double stranded DNA to access the bottom of the cleft. While further cleft closure is required to reach a transcription-competent state (*Neyer et al., 2016*; *Tafur et al., 2016*), the cleft in monomeric Pol I is about half way from inactive dimeric to elongating Pol I. Additionally, the dimer to monomer transition affects several other domains in Pol I, including movement of the jaw towards the clamp core, as well as the opening of the foot and associated subunit Rpb5 (*Figure 5D*; *Video 2*).

Our cryo-EM map of the Pol I–Rrn3 complex shows the precise location of the activating factor on the enzyme (*Figure 6A*). The elongated Rrn3 molecule binds on the concave face of a valley formed by the stalk and the dock domain in subunit A190, and extends further to reach the AC40/AC19 heterodimer at the back of the enzyme. Interestingly, Rrn3 binding fixes the Pol I stalk with respect to free monomeric Pol I, thus generating a surface for interaction with promoter-bound initiation factors (*Figure 6B*). Apart from stalk ordering, the conformation of the enzyme in the Pol I–Rrn3 complex is virtually identical to that of monomeric Pol I, indicating that conformational changes are not associated with Rrn3 binding. Comparison of our cryo-EM reconstruction with the reported structures (*Engel et al., 2016*; *Pilsl et al., 2016*) shows minor differences such as flexibility of the A12.2 C-terminal Zn ribbon and a slightly shifted orientation of Rrn3 (*Figure 6—figure supplement 1*), which reveals a certain degree of plasticity in the complex.

## The stalk subunit A14 influences rDNA promoter association

Our Pol I–Rrn3 structure underscores Pol I regions that are critical to bind the activating factor (*Figure 7A–B*). The A43 subunit strongly binds Rrn3 HEAT repeats H2-H4, through interaction surfaces of both proteins that contain several serine residues. In particular S145 in Rrn3, whose phosphomimetic mutant exhibits a growth defect (*Blattner et al., 2011*), falls at the heart of a serine cradle in the A43 OB domain formed by residues S141, S143, S156 and S244 (*Figure 7C*). The second stalk subunit, A14, contacts Rrn3 around HEAT repeat H5 using helix α2 of its tip-associated (TA) domain. The residues in this helix that are more proximal to Rrn3 include a stretch of three serines and also arginine 91 (*Figure 7D*). The central part of Rrn3 contacts both the stalk-binding domain in subunit A135 and the dock domain in subunit A190 at Pol I-specific insertions (*Figure 7E*). Finally, the C-terminal third of Rrn3 contacts subunit AC19 and the AC40 C-terminus, in agreement with data showing that K329 in AC40 crosslinks K558 in Rrn3 (*Blattner et al., 2011*).

While the relevance of A43 in Rrn3 binding has been demonstrated (*Peyroche et al., 2000*), no evidence for A14 has been reported yet. Based on our cryo-EM structure, we engineered several A14 mutants to assess their role in complex formation and promoter association. We first took advantage of the fact that cells lacking this subunit are viable to produce a knockout mutant (ΔA14). In agreement with cryo-EM data, ΔA14 cells exhibit a 75% decrease in Pol I–Rrn3 as compared to wild-type (*Figure 7F*). A similar decrease in promoter association for both components of the complex was measured by ChIP. Moreover, this strain presents a growth defect but the levels of several Pol I subunits including A43 and those of Rrn3 are unaffected (*Figure 7—figure supplement 1*). These results indicate that A14 plays a role in Pol I association to rDNA. We then produced specific deletions of different A14 structural elements that, according to our cryo-EM structure, lie in the vicinity of Rrn3 (*Figure 7*). As in the case of ΔA14, cells expressing these mutations present similar levels of A190 and Rrn3 to wild-type cells (*Figure 7—figure supplement 1B*). A strain lacking the C-terminal tail (A14ΔCt, Δ101–137) presents similar levels of Pol I–Rrn3 and promoter association as

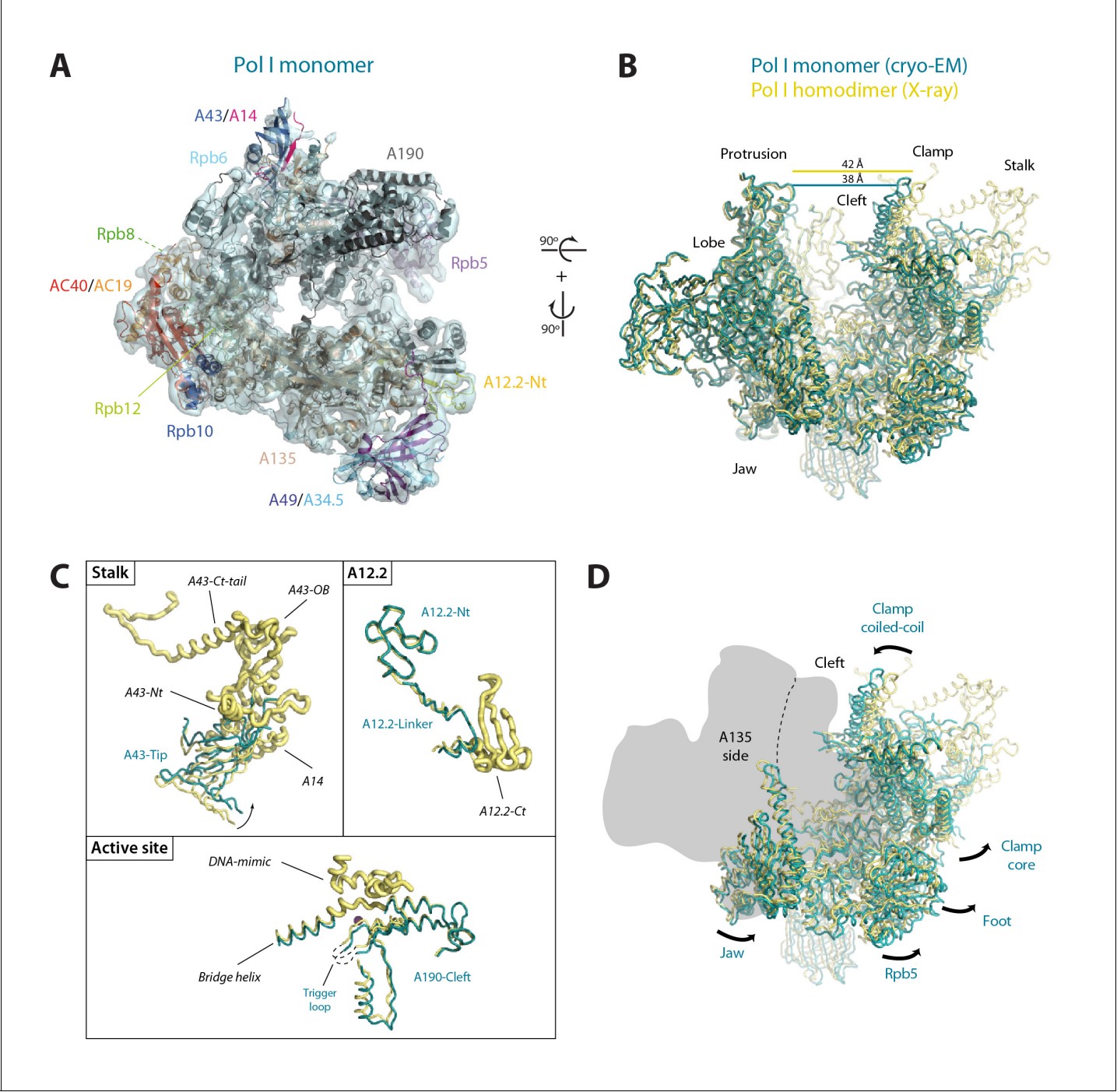

**Figure 5.** Structure of monomeric Pol I in solution. (**A**) Cryo-EM reconstruction of Pol I at 4.9 Å resolution superposed with the derived pseudo-atomic model. (**B**) Comparison between the structural models of the Pol I dimer (PDB-4C3H) and the Pol I monomer (this report) in yellow and cyan, respectively, with labelled structural domains. (**C**) Close-up views of regions becoming flexible in the transition from dimeric to monomeric Pol I, in the same colors and orientation as in panel **B**. Flexible regions are depicted with thicker ribbon trace and labelled in italics. (**D**) Representation of the conformational changes associated with the transition from dimeric to monomeric Pol I, in the same colors and orientation as in panel **B**.

The following figure supplements are available for figure 5:

**Figure supplement 1.** Negative-staining EM of the yeast Pol I–Rrn3 complex.

**Figure supplement 2.** Cryo-EM structure of the yeast Pol I–Rrn3 complex.

*Figure 5 continued on next page*

*Figure 5 continued*

**Figure supplement 3.** Structural details of the monomeric Pol I and Pol I–Rrn3 cryo-EM structures.

wild-type cells (*Figure 7F*), indicating that the A14 C-terminal tail is not involved in Rrn3 binding. In accordance, this mutant exhibits normal growth (*Figure 7—figure supplement 1A*). However, a strain that also lacks helix α2 of the TA domain (A14△α2Ct, Δ80–137) presents about one third reduction in detected Pol I–Rrn3 levels and promoter association (*Figure 7F*). While less intense, this resembles the behaviour of ΔA14, also at the growth level (*Figure 7—figure supplement 1*). More-over, we generated an A14 point mutant at arginine 91 (A14-R91E) exhibiting a reduction in detected levels of Pol I–Rrn3 and in promoter association that is comparable to that observed for ΔA14 (*Figure 7F*), while growth is less affected (*Figure 7—figure supplement 1A*). Our A14 muta-tional analysis indicates that helix α2 is fundamental for Rrn3 binding and subsequent promoter association.

Finally, we engineered a strain lacking an internal loop in the TA domain that appears disordered in the crystal structure of dimeric Pol I (A14ΔTAloop; Δ53–77). Strikingly, this mutant behaves oppo-site to ΔA14 and helix α2 mutants, as it shows a 2-fold increase in detected Pol I–Rrn3 complexes (*Figure 7F*). In accordance, we observe a 3- and 1.5-fold increase in promoter association for Pol I and Rrn3, respectively, while growth is not affected (*Figure 7F*; *Figure 7—figure supplement 1A*). When cells expressing A14ΔTAloop were cultured under starving conditions, the Pol I–Rrn3 levels are about double of the wild-type, while detected Pol I homodimers and promoter occupancy of Pol I and Rrn3 remain unaffected (*Figure 7—figure supplement 2*). This suggests that the TA-loop in A14 has a role in limiting binding to Rrn3. These data also show that higher levels of Pol I–Rrn3 are not sufficient to influence Pol I homodimerization or promoter dissociation in starved cells.

## Discussion

In this study, which includes a wealth of techniques ranging from structural analysis to in vivo interac-tion experiments, we shed light on the regulation of Pol I activation, the first step in ribosome bio-genesis. An improved PICT assay allowed us to investigate the levels of Pol I homodimers and Pol I–Rrn3 complexes in response to nutrient availability. We also provide a detailed picture of the confor-mational rearrangements taking place in the transition between the inactive and activated states of Pol I, and identify distinct stalk regions as central structural elements in this process.

### Dimerization as a Pol I storage mechanism

Cellular polymerases are recruited to the pro-moter in an active, monomeric conformation (*Hirata et al., 2008*; *Murakami et al., 2002*; *Vannini and Cramer, 2012*). However, Pol I can also form homodimers that are incompetent for transcription, yet this assembly could only be observed in vitro (*Engel et al., 2013*; *Fernán-dez-Tornero et al., 2013*; *Milkereit et al., 1997*). While we were unable to detect Pol I dimers in growing cells, nutrient depletion induced enzyme homodimerization. Moreover, these dimers also form after inhibition of either rRNA maturation with diazaborine or protein synthesis with cycloheximide. While these three processes are mechanistically different, they all

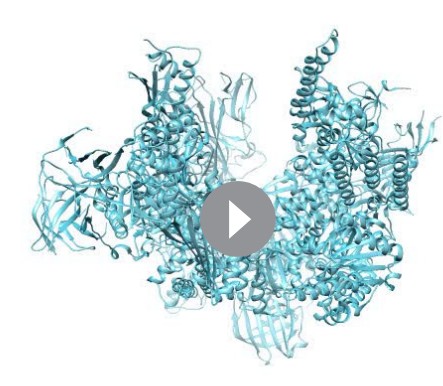

**Video 2.** Structural transition from Pol I homodimers to monomers. The Pol I enzyme is presented in the same view as *Figure 5B*, with the clamp on the right and the protrusion on the left.

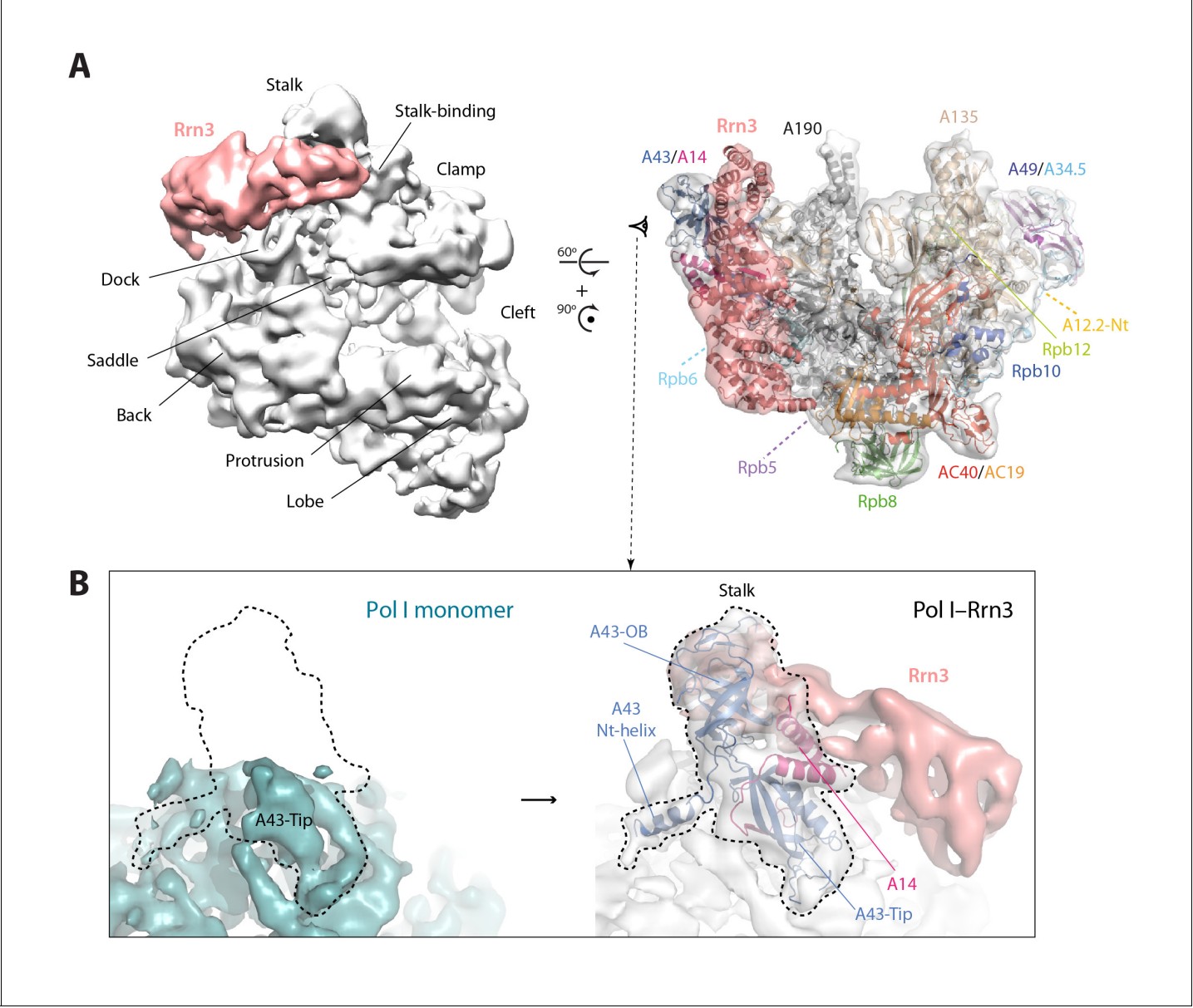

**Figure 6.** Structure of the Pol I–Rrn3 complex. (**A**) Cryo-EM reconstruction of Pol I in complex with Rrn3 (left) superposed with the derived pseudo-atomic model (right). The different Pol I structural domains and subunits are labelled. (**B**) Stalk fixation in the transition from free to Rrn3-bound Pol I, in a lateral view as indicated. Subunit A14 and the different domains in subunit A43 are indicated.

The following figure supplement is available for figure 6:

**Figure supplement 1.** Structural comparison of Pol I–Rrn3 cryo-EM structures.

negatively affect events that are downstream of Pol I transcription. It is therefore likely that, when rRNA synthesis has to be reduced, transcriptionally disengaged Pol I molecules form inactive dimers. This peculiar mechanism implies two major advantages. On one hand, the formation of compact homodimers could protect a pool of this highly-abundant enzyme from degradation, thus saving energy to the cell. On the other hand, upon restoration of favourable conditions, rRNA synthesis can be reactivated, while de novo Pol I production would delay the process, especially if few ribosomes are available. Finally, although we cannot exclude that Pol II or Pol III could homodimerize in

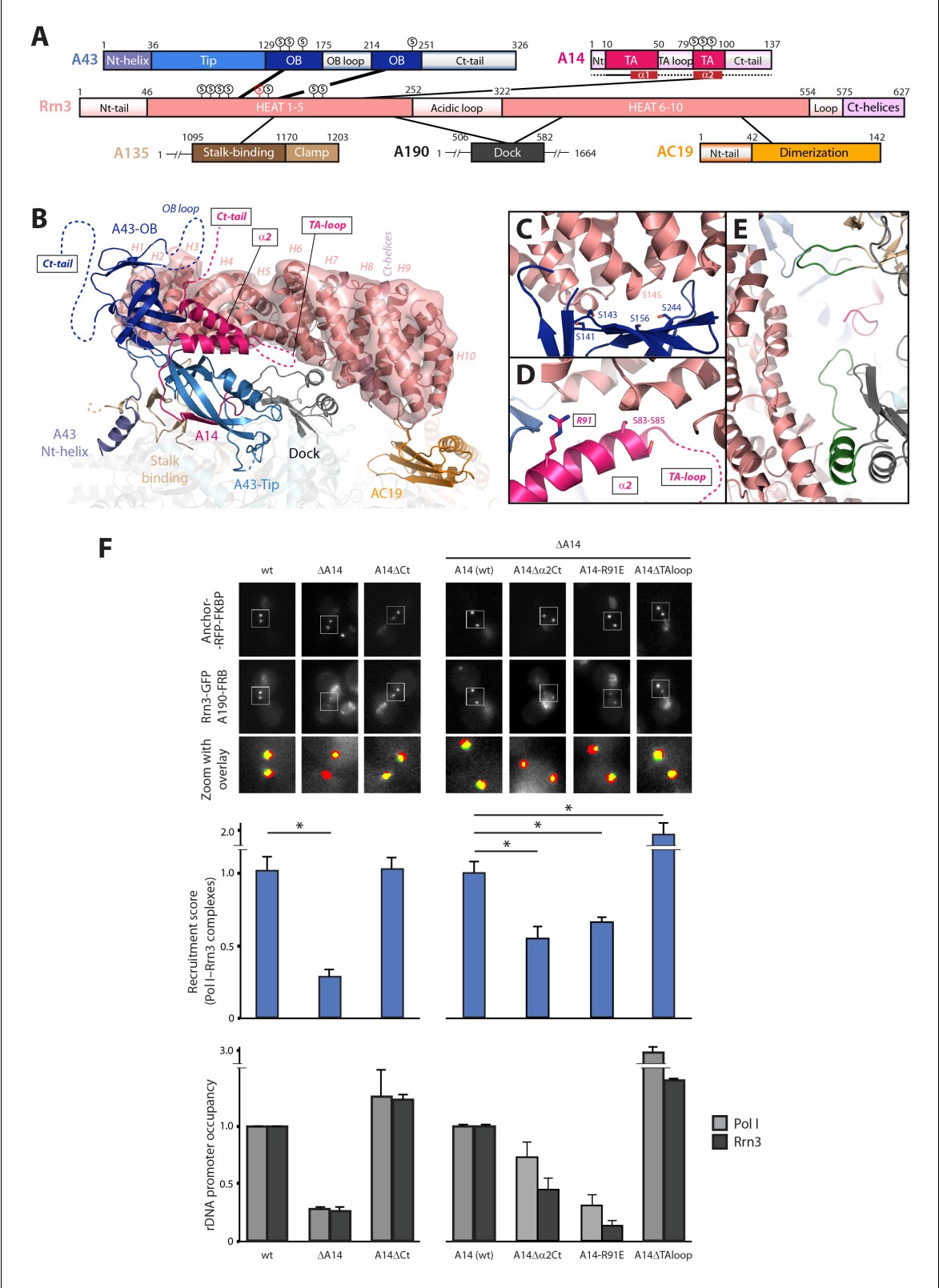

**Figure 7.** Mutational analysis of Pol I stalk subunits contacting Rrn3. (**A**) Bar diagrams of the Pol I regions involved in Rrn3 binding, with connecting lines as derived from the cryo-EM structure. α-helices 1 and 2 in the A14 TA domain are marked in dark red. Serine residues in the A43 cradle, A14 stretch and Rrn3 patch are shown above the corresponding bars, with Rrn3-S145 in red. (**B**) Close-up view of the Pol I–Rrn3 interaction in a similar orientation to that in *Figure 6B*. HEAT repeats in Rrn3 labelled H1 to H10. Dotted lines represent disordered regions in the Pol I and Rrn3 crystal structures. Boxed-

*Figure 7 continued*

text marks truncated regions in the yeast mutants of panel **F**. (**C**) Close-up view of the serine cradle in A43 that accommodates serine 145 in Rrn3. (**D**) Close-up view of A14 helix α2, which lies in the vicinity of Rrn3. (**E**) Pol I specific insertions in subunits A190 and A135 are shown in green. (**F**) Representative PICT images of the RFP-tagged anchor (upper row), GFP-tagged Rrn3 (middle row) and a zoom of a 2.6 × 2.6 μm square around anchoring platforms (bottom row) of different mutant strains. Below, quantification of the Rrn3-GFP recruitment score, normalized to the measurement of the wild-type strain (Mean ± SD, p-value * < 0.01 t-test). At the bottom, ChIP experiments showing the relative association of A190 (light) and Rrn3 (dark) to the rDNA promoter region. All ChIP experiments were normalized to the value of the wild-type strain in rich medium (Mean ± SD).

The following figure supplements are available for figure 7:

**Figure supplement 1.** Additional characterization of Pol I stalk mutants.

**Figure supplement 2.** Characterization of A14ΔTAloop in starving conditions.

conditions other than nutrient starvation, our results suggest that homodimerization is not a general mechanism to regulate eukaryotic transcription.

## Cells fine-tune the levels of Pol I complexes in response to nutrient availability

While only a minor fraction of Pol I is bound to Rrn3 in growing cells (*Milkereit and Tschochner, 1998*), the formation of this complex is a pre-requisite for transcription initiation (*Aprikian et al., 2001*; *Schnapp et al., 1993*). We show that nutrient depletion induces a rapid reduction in the levels of Pol I–Rrn3, which correlates in time with a marked decrease in the promoter association of both Pol I and Rrn3. We also show that promoter dissociation is strongly reduced in a strain where Pol I is permanently attached to Rrn3. This suggests an influence of Pol I–Rrn3 levels in rDNA transcription. However, the levels of Pol I–Rrn3 as such are not sufficient to account for transcription inactivation by nutrient depletion, as Pol I–Rrn3 complexes can be detected regardless of null promoter levels of A190 and Rrn3. This is in agreement with previous observations of rDNA transcription inactivation by inhibition of TOR signalling (*Philippi et al., 2010*). Interestingly, our A43ΔCt mutant, which specifically impairs Pol I homodimerization, shows that this mechanism also contributes to inactivate Pol I transcription in response to nutrient deprivation. Therefore, both Pol I–Rrn3 and Pol I dimerization modulate rDNA transcription, which allows us to propose a model (*Figure 8*). When nutrients are depleted, Pol I–Rrn3 levels and promoter association drop exponentially whereas Pol I only homodimerizes subsequently. Remarkably, 20 min after starvation, Pol I–Rrn3 levels remain relatively high in spite of marginal amounts of promoter association. We thus hypothesize that, in addition to Pol I–Rrn3 disassembly, additional regulatory mechanisms may contribute to initial transcriptional inactivation. At a later stage, Pol I homodimerization remains a major factor limiting transcription. Upon refeeding from starvation, we propose that available Pol I–Rrn3 complexes are rapidly recruited for transcription, while disruption of Pol I homodimers provides fresh Pol I that can interact with Rrn3 to increase Pol I–Rrn3 complexes and further activate rDNA transcription.

In contrast to eukaryotes, bacteria use a single form of RNA polymerase to transcribe their entire genome. In response to stress, bacteria induce the production of the ppGpp alarmone, which targets RNA polymerase at an allosteric site (*Mechold et al., 2013*; *Zuo et al., 2013*). This selectively destabilizes initiation complexes at GC-rich promoters, such as those of rRNA (*Travers, 1980*). The presence of a dedicated transcription system for rDNA allows eukaryotes to specifically downregulate rRNA production in order to control cell growth. Our results indicate that Pol I dimers and Pol I–Rrn3 complexes contribute to achieve this goal.

## The Pol I stalk as a sensing platform of the cell state

The yeast A43 subunit comprises an elongated core, conserved within eukaryotic RNA polymerases (*Kuhn et al., 2007*), flanked by specific N- and C-terminal extensions (*Figure 7A*). We demonstrate that the A43 C-terminal end is essential for enzyme inactivation through Pol I dimerization, in accordance with published structural data (*Engel et al., 2013*; *Fernández-Tornero et al., 2013*). The A43 C-terminal tail is conserved from yeast to human, arguing for a Pol I monomer/dimer modulation of rRNA synthesis in higher eukaryotes (*Beckouët et al., 2011*). Nevertheless, there are organisms

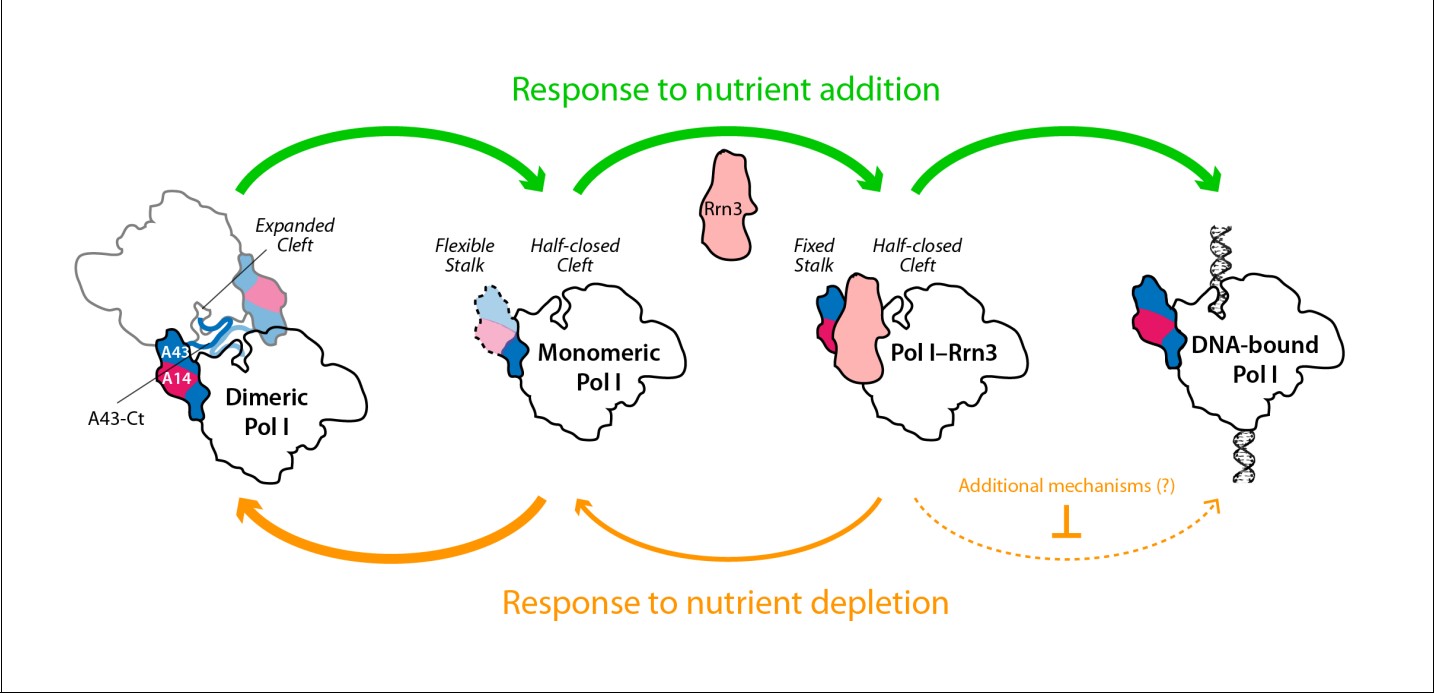

**Figure 8.** Model for the influence of nutrient availability on the assembly of Pol I complexes. Nutrient addition to starved cells induces formation of Pol I–Rrn3 complexes to activate transcription, while Pol I homodimers disrupt to generate fresh monomeric Pol I (green arrows). Nutrient depletion causes partial disruption of Pol I–Rrn3 complexes and formation of Pol I homodimers (orange arrows). Both events downregulate rDNA transcription, while additional regulatory mechanisms may also participate.

The following figure supplement is available for figure 8:

**Figure supplement 1.** Comparison with other transcription systems.

lacking the A43 C-terminal tail, such as *S. pombe* and *A. thaliana*, where likely no Pol I dimerization will take place in nutrient-arrested cells. Additionally, our cryo-EM structure shows that the main Pol I element involved in Rrn3 interaction is the OB domain in the A43 elongated core. This is consistent with reported data in yeast (*Peyroche et al., 2000*) and with the identification of an OB peptide that can bind Rrn3 in human (*Rothblum et al., 2014*). Moreover, our structure points towards a contact between a serine cradle in A43 and a serine patch in Rrn3. In agreement, it was shown that this interaction can only take place if Pol I is phosphorylated and Rrn3 is not (*Fath et al., 2001*; *Gerber et al., 2008*).

Additionally, we show that A14 is also involved in Rrn3 interaction. When A14 is deleted, both Rrn3 binding and promoter association of both Pol I and Rrn3 are severely impaired. A similar behaviour can be obtained by either deletion of the last 58 residues in subunit A14 or the A14-R91E point mutant. This suggests that A14, and helix α2 in particular, plays an important role in Pol I–Rrn3 complex formation, as suggested by cryo-EM data. Furthermore, the TA-loop, which is disordered in the dimeric Pol I crystal structure, is involved in limiting Rrn3 binding to Pol I. Deletion of this region leads to a strong increase in both Pol I–Rrn3 complex formation and its association with the rDNA promoter in growing cells. Genetic studies in *S. pombe* implicated the A14 homolog in Rrn3 binding (*Imazawa et al., 2005*), supporting that our A14 observations may be extended to other organisms.

## A common surface for activating cofactors

Three structural elements in Pol I are involved in the interaction with Rrn3: (i) the stalk, (ii) the dock domain, and (iii) the AC40/AC19 heterodimer. Contact of the activating factor with these three regions is therefore required for activation and subsequent binding to promoter-attached initiation factors. In the Pol II system, the Mediator complex acts as an activating cofactor required for

association of the enzyme to promoter-bound initiation factors (*Kim et al., 1994*). Interestingly, the Pol II–Mediator structure shows that the coactivator contacts the enzyme at the stalk, the dock domain, and the AC40/AC19 homolog (*Plaschka et al., 2015*). While other details of the enzyme-cofactor interaction differ between both transcription systems (*Figure 8—figure supplement 1A*), the described similarities suggest that the interaction of activating factors with the stalk and surrounding regions might help bring RNA polymerases to the promoter. Additionally, in spite of significant differences with bacterial transcription, the σ factor may also be regarded as an activator, as its binding is required for promoter association. Fitting of the bacterial holoenzyme structure into our cryo-EM map shows that domain 4 of the σ factor ($\sigma^4$) falls in the region where Rrn3 is located and contacts equivalent domains of the RNA polymerase core (*Figure 8—figure supplement 1B*). Moreover, $\sigma^4$ binds the bacterial promoter DNA (*Murakami et al., 2002*) and mammalian Rrn3 was shown to interact with the rDNA promoter (*Stepanchick et al., 2013*). These analogies suggest that an overall architectural arrangement may be conserved to mediate enzyme activation and promoter recruitment.

## Materials and methods

### Yeast strains

The strains used are listed in *Supplementary file 1*. Strain construction and other genetic manipulations were performed following standard procedures (*Burke et al., 2000*). Briefly, FKBP-RFP, FRB and GFP tagging was performed by gene replacement using standard PCR strategies. The same was done for deletions of *RPA14* (*rpa14Δ*, *ΔA14*) and the C-terminal regions of *RPA14* (A14ΔCt) and *RPA43* (A43ΔCt). The remaining A14 mutants were obtained by directed mutagenesis of the wild type *RPA14* gene cloned in a centromeric plasmid, which was used to transform the *rpa14Δ* strain, followed by selection in appropriate medium.

### Live-cell imaging

The PICT assay was carried out as described (*Gallego et al., 2013*). For analysis, strains were grown in YPD medium at 30°C overnight, then diluted and grown up to exponential phase in synthetic complete (SC) medium. Cells attached to 35 mm glass bottom culture dishes coated with Concanavalin A were incubated for 2 hr at 30°C in either SC-Low Fluorescence medium (lacking folic acid and riboflavine) or the same medium without nitrogen and glucose. Where indicated, 10 µM rapamycin was also added, either alone or together with 0.2 µg/ml cycloheximide or 10 µg/ml diazaborine. Images were acquired on an Olympus IX81 microscope equipped with 100x/1.30 objective lens, a Hamamatsu Orca-ER camera, and two complete fluorescence filter cubes from AHF respectively optimized for GFP (ET Bandpass 470/40 + Beamsplitter 500DVXRUV + ET Bandpass 525/50) and RFP (ET Bandpass 545/30 + Beamsplitter 580 DVXRUV + BrightLine HC 617/73). All strains were analyzed in three biological replicates, where each sample was imaged in at least 6 fields of view close to the equatorial section of the cells. The acquisition of both fluorescence channels was performed sequentially by switching the filter cubes. The software ImageJ (http://rsb.info.nih.gov/ij/) was used to analyze the images and a custom image analysis workflow was developed and implemented in ImageJ macro language to enable the automatic processing of complete data sets. This workflow independently processes RFP (red) and GFP (green) channels to segment spot like structures of a specific size in the images; the functional steps (*Figure 1—figure supplement 2*) are identical for both channels but the settings are slightly different to adapt to the difference in image quality. The recruitment score is defined as the ratio of (i) the summed area of all segmented prey-GFP spots overlapping with segmented RFP-anchors multiplied by the mean GFP intensity measured inside this region, to (ii) the summed area of all segmented RFP-anchors. The overall recruitment score for a condition is hence a measurement performed over tens of small image regions coming from all the images of a given replica. The segmentation of RFP-anchors is almost flawless since the signal to noise ratio is very high for this channel. Prey-GFP spots are more challenging to segment due to their lower contrast and the presence of a strong but smoothly varying background signal, still the results are reasonably good thanks to the pre-filtering and local thresholding adapted to the expected spots geometry, and the automatic validation of the segmented particles based on their geometry (*Figure 1—figure supplement 2*). Furthermore, from the recruitment score definition, only false positive

prey-GFP spots overlapping with RFP-anchors do actually count toward the recruitment score; since anchors are sparsely located, and are virtually segmented flawlessly, this only happens very marginally. Finally, no condition specific increase in GFP signal background intensity could be detected by our control experiments (*Figure 1—figure supplement 3F*; *Figure 3—figure supplement 1B*). Overall, the recruitment score is hence highly reproducible, which is backed by the low standard deviations observed across the three biological replicates of each condition.

Localization of the recruited prey-GFP on the Tub4-RFP-FKBP anchoring platform was done on images obtained from the corresponding PICT assay. Images were background subtracted and cells with two anchoring platforms were selected. Intensity profile was extracted for a line that linked the brightest pixel of each Tub4-RFP-FKBP anchoring platform, both for the red and the green channel. The ImageJ software (http://rsb.info.nih.gov/ij/) was used to analyze 10 cells for each strain. As the nucleus is found between two Tub4-RFP-FKBP anchoring platforms of yeast cells with two spindle pole bodies, recruited preys-GFP that specifically accumulate between the two anchoring platforms were scored as nuclear (*Figure 1—figure supplement 1B*).

## Co-immunoprecipitation (co-IP)

Cells were grown to early logarithm phase and then half of the culture volume was crosslinked while the remaining half was filtered, washed, and incubated in starving medium for 2 hr. The latter was then crosslinked after the 2 hr incubation. After crosslinking the two halves were harvested, washed with water, and suspended in lysis buffer containing protease and phosphatase inhibitors. The cell suspension was flash frozen in liquid nitrogen, and then ground to a fine powder using a chilled mortar. Afterwards, the cell lysate was thawed slowly on ice, transferred to pre-chilled tubes and centrifuged. The supernatant was collected and total protein concentration was estimated. The pellet containing the chromatin insoluble fraction was resolubilized in lysis buffer by sonication. After clarification by centrifugation, the supernatant was recovered. A190-TAP was then precipitated, the volume of each cell extract containing 20 mg of protein was incubated with 50 µl of IgG Sepharose 6FF (GE Healthcare, Pittsburgh, PA) slurry overnight at 4°C, and then, after extensive washing and decrosslinking at 65°C for several hours, analysed by western-blot using a monoclonal antibody against the Myc epitope.

## Chromatin immunoprecipitation (ChIP)

ChIP and purification, quantitative real-time PCR (qPCR) amplification and data analysis were performed described (*García et al., 2010*). For starvation experiments, cells were grown to early logarithm phase and then half of the cultures were crosslinked and half harvested, washed and incubated in starvation media for 1 hr. Then, the second half cultures were crosslinked. Following ChIP and purification, qPCR was performed with a CFX96 Detection System (Bio-Rad, Hercules, California), using SsoAdvanced Universal SYBR Green Supermix (Bio-Rad) following manufacturer's instructions. Anti-GFP for Rrn3 analysis and anti-A190 are from rabbit (Molecular Probes and gift of Michel Riva, respectively). Four serial 10-fold dilutions of genomic DNA were amplified, using the same reaction mixture as for samples to construct the standard curves. qPCR reactions were performed in triplicate and with at least three independent ChIPs (biological replicate). Each biological replicate contained two technical replicates. Quantitative analysis was carried out using the CFX96 Manager Software (version 3.1, Bio-Rad). Plotted data correspond to mean values from at least three different experiments and the error bars represent standard deviations. To characterize the rDNA occupancy of Pol I subunit A190 and Rrn3-GFP, a 35S rDNA promoter region and a 5S rRNA gene region were amplified. After normalizing the IP to the respective input values and to a non-transcribed region of chromosome VII, relative occupancies were obtained by relating data from the promoter region to the 5S rDNA as described (*Philippi et al., 2010*).

## Protein purification and Pol I–Rrn3 complex formation

Rrn3 was amplified by PCR from yeast genomic DNA and cloned into pETM11 between NcoI and Acc65I restriction sites, resulting in an N-terminal hexahistidine tag followed by a tobacco etch virus (TEV) protease target sequence. His-TEV-Rrn3 was expressed in *E. coli* Rosetta Cells (Novagen, Madison, Wisconsin) in TB autoinducing medium overnight at 24°C. Cells were harvested by centrifugation and resuspended in L Buffer (50 mM HEPES pH 7.8, 200 mM NaCl, 10% glycerol, 15 mM

imidazol, 2 mM beta-mercaptoethanol) supplemented with protease inhibitors (cOmplete EDTA-free, Roche, Switzerland). Cells were sonicated and the lysate was centrifuged at 20,000 g for 40 min. The supernatant was loaded on HisTrap (GE Healthcare) equilibrated in L Buffer with 0.5 mM Phenylmethylsulfonyl fluoride and eluted in a linear gradient to 400 mM imidazole. Rrn3-containing fractions were pooled, loaded on Mono Q (GE Healthcare) equilibrated in MQ Buffer (50 mM HEPES pH 7.8, 200 mM NaCl, 5 mM DTT), and eluted in a linear gradient to 750 mM NaCl. Pooled fractions were concentrated and loaded on a Superdex 200 (GE Healthcare) equilibrated in GF Buffer (20 mM HEPES pH 7.8, 100 mM $Na_2SO_4$, 1 mM $MgCl_2$, 10 μM $ZnCl_2$, 5 mM DTT). Eluted fractions were concentrated to 20 mg/mL, frozen in liquid $N_2$ and stored at $-80°C$.

Pol I was obtained according to published protocols (*Moreno-Morcillo et al., 2014*). A Pol I variant lacking the last 49 residues in subunit A43 was isolated from standard purifications, as a distinct peak in ionic exchange chromatography. For Pol I–Rrn3 complex preparation, the enzyme was dialyzed against GF Buffer and incubated with Rrn3 in a 1:1 molar ratio on ice for 16 hr. The sample was cross-linked with 0.16% glutaraldehyde for 5 min on ice and subsequently quenched with 25 mM Tris pH 8.3, 200 mM glycine. This sample was injected in a Superdex 200 (GE Healthcare) size exclusion column equilibrated in EM buffer (20 mM HEPES pH 7.8, 100 mM NaCl). The quality of the crosslinked sample was assessed by SDS-PAGE, native gel electrophoresis and LC-ESI MS/MS. For Rrn3 antibody labelling in the context of the Pol I–Rrn3 complex, Pol I was incubated with Rrn3 in a 1:1 molar ratio overnight at 4°C in GF buffer, supplemented with HEPES up to 110 mM. The sample was crosslinked by incubation with 0.48% glutaraldehyde for 5 min on ice and then quenched with 25 mM Tris pH 8.3, 200 mM glycine. The StrepMab-Immo antibody (IBA, Germany) was then added to form the Pol I–Rrn3–antibody complex (1:1:1.5 molar ratio). The mixture was loaded in a Superose 6 (GE Healthcare) equilibrated in GF1 buffer (20 mM HEPES pH 7.8, 85 mM $Na_2SO_4$, 3 mM DTT) to isolate the complex of interest from the free antibody and crosslinking aggregates.

## Electron microscopy

For negative-staining EM, crosslinked Pol I–Rrn3 (20 ng/μl) was adsorbed on a glow-discharged carbon-coated copper grid and stained with 2% uranyl formate. Observations were performed in a JEOL-1230 electron microscope operated at 100 kV and micrographs were recorded under low-dose conditions (~10 e–/Å2) using a 4k × 4k TemCam-F416 camera (TVIPS) at 2.28 Å/pixel. The same procedure was used for the Pol I–Rrn3–antibody complex, except that micrographs were collected at 2.84 Å/pix.

For cryo-EM, 4 μl of crosslinked Pol I–Rrn3 (60 ng/μl) were placed onto glow-discharged Quantifoil R2/2 grids and incubated in the chamber of a FEI Vitrobot at 4°C and 95% humidity for 30 s before blotting for 2 s at an offset of –2 mm. Data were collected on a FEI Titan Krios electron microscope operated at 300 kV, using FEI automated single particle acquisition software (EPU) on a back-thinned FEI Falcon II detector at a calibrated magnification of 79,096 (pixel size of 1.77 Å). Defocus values ranged from 1.9 to 4.2 μm. Videos were intercepted at a rate of 68 frames for 4 s exposures.

## Image processing

For negative-staining, around 63,500 images of Pol I and 42,400 images of Pol I–Rrn3 were extracted using EMAN (*Tang et al., 2007*) and binned to 5.68 Å/pixel of 2D analysis. The contrast transfer function (CFT) was estimated using CTFFIND3 (*Mindell and Grigorieff, 2003*) and corrected by flipping phases. Reference-free averages were obtained using EMAN, while 3D reconstructions were carried out using protocols implemented in Xmipp (*Scheres et al., 2008*). The correctness of the final structures was supported by the high correlation between 2D projections of the models and reference-free averages. In the case of Pol I–Rrn3–antibody, data processing was done in Scipion (*de la Rosa-Trevín et al., 2016*) following a similar procedure except that 10,997 particles selected after 2D classification were subjected to several rounds of 3D classification using Relion (*Scheres, 2012*). A 3D class containing 1355 particles clearly corresponded to Pol I–Rrn3–antibody complex.

For cryo-EM, 1288 movies were averaged using optical flow correction (*Abrishami et al., 2015*) and their CTF was estimated using CTFFIND4 (*Mindell and Grigorieff, 2003*). Approximately 230,000 particles were automatically selected using RELION (*Scheres, 2012*), also employed for subsequent data processing (*Supplementary file 2*). Two rounds of reference-free 2D class averaging

allowed removal of bad particles, yielding a stack of 190,750 good-quality images. A negative-staining reconstruction of free Pol I was low-pass filtered at 60 Å and used as starting model to sort the images using 3D classification. Particles were split into six classes with T = 4, an offset search range of 6 pixels, and offset search steps of 2 pixels. Only one class containing a total of 32,175 images clearly showed density for both Pol I and Rrn3, while a second class with 90,173 particles corresponds to monomeric Pol I. Both particle sets were subjected to 3D refinement including particle polishing and post-processing, which yielded maps for monomeric Pol I and Pol I–Rrn3 with final resolutions of 5.6 and 7.7 Å, respectively, according to the gold-standard FSC = 0.143. Finally, both particle sets were added together and subjected to the same refinement procedure, producing a map with a resolution of 4.9 Å.

### Structure modeling

The available crystal structure of dimeric Pol I (PDB entry 4C3H) was fitted into the cryo-EM map of monomeric Pol I at 4.9 Å resolution, using UCSF Chimera (*Pettersen et al., 2004*). Regions that appeared disordered in the cryo-EM map were deleted from the model using COOT (*Emsley and Cowtan, 2004*). The resulting model was divided into 30 different rigid bodies, as previously defined (*Moreno-Morcillo et al., 2014*), subjected to rigid-body real-space refinement using PHENIX (*Adams et al., 2010*) and finally corrected for chain breaks. The same procedure was used for the cryo-EM map of the Pol I–Rrn3 complex by the addition of the Rrn3 crystal structure (PDB entry 3TJ1), which was divided into two rigid bodies at the disordered acidic loop located in the middle region of the Rrn3 structure. Figures were prepared with UCSF Chimera or Pymol (www.pymol.org).

### Data availability

The Pol I–Rrn3 and Pol I monomer cryo-EM maps were deposited under accession numbers EMD-4086 and EMD-4087. The cryo-EM map of the Pol I monomer at 4.9 Å resolution and its corresponding pseudo-atomic model were deposited under accession codes EMD-4088 and PDB-5LMX.

## Acknowledgements

The authors would like to thank KS Murakami and M Moreno-Morcillo for critically reading the manuscript. We are also grateful to IS Fernández, S Scheres, I Fita, R Méndez, KR Vinothkumar and NMI Taylor for helpful advice. We express our gratitude to M Riva and O Gadal for providing plasmids, strains and antibodies, the LMB-MRC for access to electron microscopes, V Abrishami for movie processing, D Ureña for yeast growth, H Bergler for the diazaborine compound, A Paradela for mass spectrometry, the ADMCF-IRB for live-cell imaging, and the CIB Protein Chemistry Facility for N-terminal sequencing. The project was supported by grant BFU2013-48374-P of the Spanish MINECO and by the Ramón Areces Foundation. OG held a research contract under the Ramón y Cajal program of the Spanish MINECO (RYC-2011–07967). IRB Barcelona is the recipient of a Severo Ochoa Award of Excellence from the Spanish MINECO.

## Additional information

### Funding

| Funder | Grant reference number | Author |
|---|---|---|
| Ministerio de Economía y Competitividad | BFU2013-48374-P | Carlos Fernández-Tornero |
| Fundación Ramón Areces | | Carlos Fernández-Tornero |
| Ministerio de Economía y Competitividad | RYC-2011-07967 | Oriol Gallego |

The funders had no role in study design, data collection and interpretation, or the decision to submit the work for publication.

## Author contributions

ET, Conceptualization, Formal analysis, Validation, Investigation, Methodology, Writing—review and editing, Designed, carried out and analyzed protein complex preparation and electron microscopy experiments, and assisted with reviewing/editing the manuscript; JAL, Formal analysis, Investigation, Methodology, Writing—review and editing, Designed, carried out and analyzed protein complex preparation and electron microscopy experiments, and assisted with preparing the manuscript; IP, Formal analysis, Investigation, Methodology, Designed, carried out and analysed PICT experiments and constructed yeast strains; NG-P, Investigation, Constructed yeast strains, and carried out ChIP, CoIP, western blot and growth phenotype analysis; DG-C, Investigation, Methodology, Carried out cryo-EM grid preparation and quality control; AGD, Investigation, Methodology, Developed the highly sensitive PICT assay and constructed yeast strains; ST, Formal analysis, Methodology, Writing—review and editing, Designed and analysed PICT experiments and contributed to prepare the manuscript; OG, Conceptualization, Resources, Formal analysis, Supervision, Validation, Investigation, Methodology, Writing—review and editing, Developed the highly sensitive PICT assay, designed and analyzed PICT experiments, and contributed to prepare the manuscript; OC, Conceptualization, Resources, Formal analysis, Supervision, Validation, Investigation, Methodology, Writing—review and editing, Designed, carried out and analyzed ChIP experiments and CoIP assays, constructed yeast strains and contributed to prepare the manuscript; CF-T, Conceptualization, Resources, Formal analysis, Supervision, Funding acquisition, Validation, Investigation, Methodology, Writing—original draft, Project administration, Writing—review and editing, Designed, carried out and analyzed protein complex preparation and electron microscopy experiments, contributed to design all other experiments, wrote the paper, and supervised the project

## Author ORCIDs

Jaime Alegrio Louro, http://orcid.org/0000-0002-2800-923X
Carlos Fernández-Tornero, http://orcid.org/0000-0001-5097-731X

# Additional files

## Supplementary files

• Supplementary file 1. Table of yeast strains used in this study.

• Supplementary file 2. Table of statistics for the cryo-EM structures described in this study.

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
