## [Decision Letter]

Thank you for submitting your article "The dynamic balance between RNA polymerase I complexes regulates rDNA transcriptional activation" for consideration by *eLife*. Your article has been favorably evaluated by Kevin Struhl (Senior Editor) and three reviewers, one of whom, Alan G Hinnebusch (Reviewer #1), is a member of our Board of Reviewing Editors. The following individuals involved in review of your submission have agreed to reveal their identity:.

The reviewers have discussed the reviews with one another and the Reviewing Editor has drafted this decision to help you prepare a revised submission.

Summary:

In this paper, the authors study the down-regulation of Pol I activity by nutrient starvation, focusing on the roles of Pol I homodimerization and dissociation of the activated complex formed with cofactor Rrn3 as possible mechanisms that diminish recruitment of the Pol I:Rrn3 complex to the RDN promoter. Introducing an improvement in the PICT assay, they provide the first evidence that Pol I forms homodimers in cells, in a manner that depends on nutrient starvation or inhibition of either translation or ribosome biogenesis using drugs. The dimerization requires the C-terminal domain of subunit A43, implicated in dimerization by the previous crystal structure of a Pol I dimer. They could also use the PICT assay to monitor Pol I:Rrn3 complex levels and show that these are down-regulated in starved cells, although not to the same extent as promoter occupancy of these proteins measured by ChIP. They examine the kinetics of formation/dissociation of the Pol I dimer and Pol I:Rrn3 complexes by PICT in the transition from growth to starvation and back to growth, which are complex and multiphasic and show that at least to some extent the two processes are not tightly coupled in time. They then switch gears and present two cryo-EM structures, a Pol I::Rrn3 complex at 7.7 Å resolution, which appears similar in most important respects to a higher resolution structure (4.8 Å) of the same complex recently published by another group; and a Pol I monomer at 4.9 Å, of higher resolution than one also published this year. Comparison of their Pol I monomer to the crystallized dimer gives some insights into the transition from the presumably inactive dimer to active monomer, but these transitions were either not dramatic or were not well explained in the text/figures. The Pol I:Rrn3 complex is highly relevant to Pol I activation and they go on to perform a limited structure-function analysis of this interaction, showing that it is impaired if the A14 subunit (which interacts extensively with Rrn3) is simply deleted from Pol I. The Pol I:Rrn3 complex assembly and recruitment are either diminished or enhanced if the unstructured C-terminal tail of A43 is removed or the internal unstructured "TA-loop" of A14 is removed, respectively. The effect of deleting A14 entirely could be indirect, as it also interacts extensively with A43; and the opposing effects of deleting A43-CTT or A14 TA-loop, while interesting, are difficult to rationalize because they are unstructured segments that don't make known contacts with Rrn3.

Essential revisions:

The consensus of the reviewers is that the structural information in the second half of the paper is valuable, even if mostly confirmatory of other recently published structures, as are the structure-function studies, even if somewhat limited in scope. However, the structural insights into the transition from presumably inactive dimer to active monomer gleaned from the structures were either not dramatic or not well explained in the text/figures and should be highlighted better. In addition, the possible effects of the A14 subunit deletion on Pol I integrity and abundance would have to be examined.

Regarding the results obtained using the PICT assay in the first part of the paper, there are important concerns about whether the results are sufficient to conclude, as the title of the paper implies, that formation/disassembly of the homodimer versus Pol I-Rrn3 complexes are important events regulating rDNA transcription in response to nutrient availability. All three reviewers had difficulty understanding how the PICT results have been quantified; Rev. #2 had the most serious concerns, and has explained the difficulties thoroughly in his/her review. The large halo of background GFP signal compared to the small amount associated with the tethered RFP makes quantification of the co-localizing PICT signal highly questionable; and the images in Figure 1 and Figure 2 raise doubts about whether the degree of co-localization of the anchor and Pol I-GFP signals can be reliably measured. The PICT homodimerization data seem to indicate that only a small fraction of the Pol I exists in homodimers, and the Pol-Rrn3 PICT data indicate only a modest reduction in this complex (which might merely reflect reduced Rrn3 abundance) despite a strong loss of Rrn3/Pol I occupancy of the RDN promoter measured by ChIP. The analysis of the kinetics of forming/dissociating these complexes by PICT do not indicate tight coordination of Pol-Rrn3 dissociation and homodimer formation, and provide no evidence that homodimerization is driving dissociation of Pol I-Rrn3 complexes. The rates of forming/dissociating these complexes measured by PICT are quite slow and have not been compared to the rates of Rrn3/Pol I association/dissociation from the promoter measured by ChIP. As such, formation/dissociation of these protein complexes may be secondary to the key events that regulate promoter occupancy at much higher rates. In addition, the conclusions that inactive ribosomes trigger Pol I dimerization are not substantiated by the data and represent an overinterpretation of the data. Thus, while the PICT assay might be providing evidence for Pol I dimerization in cells that is triggered by nutrient starvation, provided the criticisms raised by Rev. #2 can be satisfied, it is unclear that dimerization is extensive enough and occurs with the proper kinetics to represent an important mechanism for down-regulating Pol I function, as opposed to a secondary event that might simply protect idle Pol I molecules from degradation. One way to support your claim would be to construct a mutation in the A43-CTT that specifically impairs homodimerization and show that this mutation abolishes homodimerization by PICT and also dampens loss of Pol 1:Rrn3 promoter complexes in starved cells measured by ChIP.

Specific recommendations:

1) It would be very helpful if they had an independent assay(s) for dimerization. Two choices. First, they could formaldehyde crosslink under the various in vivo situation and do a co-IP with essentially the same constructs they already have. Second, they could replace the targeting tags with Gal4, which would allow Gal4-Pol I fusions to targeted to Gal4 binding sites (assayed by ChIP). If the model is correct, one should see the non-Gal4 version of Pol I associating with Gal4 binding sites under starvation. Both of these approaches might be better for kinetics and quantification, but the real virtue of them is that they are independent assays for dimerization.

2) The authors haven't demonstrated "regulation", so their title needs to be softened. What they have "shown" is that the dimerization and change in complexes is associated with regulation.

*Reviewer #1:*

Summary of work:

In this paper, the authors study the down-regulation of Pol I activity by nutrient starvation, focusing on the roles of Pol I homodimerization and dissociation of the activated complex formed with cofactor Rrn3 as possible mechanisms that diminish recruitment of the Pol I:Rrn3 complex to the RDN promoter. Introducing an improvement in the PICT assay, they provide the first evidence that Pol I forms homodimers in cells, in a manner that depends on nutrient starvation or inhibition of either translation or ribosome biogenesis using drugs. The dimerization requires the C-terminal domain of subunit A43, implicated in dimerization by the previous crystal structure of a Pol I dimer. They could also use the PICT assay to monitor Pol I:Rrrn3 complex levels and show that these are down-regulated in starved cells, although not to the same extent as promoter occupancy of these proteins measured by ChIP. They examine the kinetics of formation/dissociation of the Pol I dimer and Pol I:Rrn3 complexes by PICT in the transition from growth to starvation and back to growth, which are complex and multiphasic and show that at least to some extent the two processes are not tightly coupled in time. They then switch gears and present two cryo-EM structures, a Pol I::Rrn3 complex at 7.7 Å resolution, which appears to be similar in all important respects to a higher resolution structure (4.8 Å) of the same complex recently published by another group; and a Pol I monomer at 4.9 Å, of higher resolution than one also published this year. Comparison of their Pol I monomer to the crystallized dimer gives some insights into the transition from the presumably inactive dimer to active monomer, but these transitions were either not dramatic or were not well explained in the text/figures. The Pol I:Rrn3 complex is highly relevant to Pol I activation and they go on to perform a limited structure-function analysis of this interaction, showing that it is impaired if the A14 subunit (which interacts extensively with Rrn3) is simply deleted from Pol I. the Pol I:Rrn3 complex assembly and recruitment are either diminished or enhanced if the unstructured C-terminal tail of A43 is removed or the internal unstructured "TA-loop" of A14 is removed, respectively. The effect of deleting A14 entirely could be indirect, as it also interacts extensively with A43, and the opposing effects of deleting A43-CTT or A14 TA-loop, while interesting, are difficult to rationalize because they are unstructured segments that don't make known contacts with Rrn3. Unfortunately, they did not make any "surgical" alterations to A14 or A43 at points of direct contact with Rrn3 in an attempt to validate the physiological relevance of the cryoEM structure for understanding assembly of the Pol I:Rrn3 complex.

General critique:

Previous work on Pol I in yeast has shown that Rrn3 binding to the enzyme is required for Pol I recruitment to the promoter, and that Pol I can form homodimers in solution and was crystallized as a homodimer with structural features indicating an inactive conformation. This paper uses the PICT assay to provide convincing evidence that Pol I homodimers can form in cells, but only in stress conditions including starvation for carbon and nitrogen, arrest of translation elongation by cycloheximide, or treatment with a drug shown to impair ribosome biogenesis. The evidence that homodimerization occurs in cells is firm, and the fact that it was observed only in starved/stressed cells supports its role as a regulatory mechanism for inactivation of Pol I under conditions where new ribosome synthesis is unwanted. Unfortunately, however, there is no direct evidence establishing its importance in down-regulating Pol I function. In fact, the kinetics of homodimerization versus PolI:Rrn3 interaction shown in Figure 4 reveal that the initial, and most rapid phase of dissociation of Pol I:Rrn3 on the shift from growth to starvation precedes any appreciable formation of Pol I dimers, and hence, is clearly not being driven, at least initially, by dimerization. What is needed in my view is a mutation that specifically impairs homodimerization, such as in the A43-CTT, and to show that this mutation would dampen the loss of Pol 1:Rrn3 complexes from promoter DNA in starved cells, thereby showing that dimerization is an important component of the down-regulation of Pol I recruitment/transcription. Unfortunately, the mutation they examined that simply deletes the entire A43-CT has the complication of also, inexplicably, impairing Pol I:Rrn3 interaction and promoter recruitment. Thus, a more surgical mutational approach would be needed to achieve this important goal.

The PICT assay also gives evidence of reduced Pol I:Rrn3 association in starved cells by ~50%, and there is an even greater reduction in Pol I/Rrn3 occupancy of the Pol I promoter measured by ChIP. However, the Western in Figure 1—figure supplement 2 shows reduced Rrn3 abundance that could be as much as 50%, which would lead to a different and less interesting interpretation of the PICT data, with no evidence for a weaker Pol I:Rrn3 association occurring in starved cells.

The cryo-EM analysis of the Pol I monomer at 4.9 Å resolution and the Pol I::Rrn3 complex at 7.7 Å resolution are valuable, although another group recently published a Pol I-Rrn3 complex at higher resolution (4.9 Å) that seems to contain all of the same key features and allows the same insights into this important intermediate in Pol I activation as those described here-thus diminishing the significance of the current structure of the complex. The higher resolution of their monomeric Pol I compared to that recently published by another group, should allow a superior comparison between the monomeric and dimer Pol I (from the crystal structure), from which they conclude that the transition from dimer to monomer "involves partial cleft closure and increased flexibility of critical motifs". However, the relevant figure (Figure 5) does not make a convincing case for the cleft closure in the monomer, and the accompanying text (subsection “Cryo-EM structures of monomeric Pol I alone and in complex with Rrn3”) did not help much to convince me. In addition, it seems possible that the critical motifs are flexible in both the monomer and dimer but were fixed in the dimer by crystal contacts.

The experiments showing that deleting the A14 subunit weakens Pol I:Rrn3 interaction by PICT assay and reduces Pol I/Rrrn3 occupancy at the promoter by ChIP assay are valuable, but they have not ruled out the possibility that removing this subunit perturbs other Pol I subunits, e.g. A43 with which it seems to interact with extensively, and thus impairs Pol I interaction with Rrn3 indirectly by disrupting A43:Rrn3 contacts rather than eliminating the A14:Rrn3 interactions evident in the cryoEM structure. It is even possible that the steady state level of Pol I is reduced by the A14 deletion and contributes to the reduced yield of Pol I:Rrn3 interactions and Pol I recruitment. Thus, the effects of the A14 subunit deletion on Pol I integrity and abundance should have been examined. But even more importantly, the authors should have used the molecular details of the cryoEM structure to more surgically disrupt the A14:Rrn3 interaction by truncating or substituting helix 2 of A14 in a way designed not to disrupt A14 interactions with other Pol I subunits. This seems like a missed opportunity to exploit their structure.

In summary, the paper represents a collection of three different lines of work joined by the theme of regulating Pol I:Rrn3 interaction. The evidence for homodimer formation in cells is interesting and valuable but it is unclear that it constitutes an important means of impeding Pol I:Rrn3 association and promoter recruitment rather than being a byproduct of the loss of this complex by other means that serves only to protect Pol I from degradation. The cryoEM structures are valuable, but a Pol I:Rrn3 complex of higher resolution was already published. The structure-function analysis of A43 and A14 did not exploit the cryoEM structure, as it involved deletions of large segments that are not present at their interfaces with Rrn3. The result that deleting the TA loop in A14 leads to greater Pol I:Rrn3 association and promoter occupancy is the most interesting finding, but the molecular mechanism is not obvious from the structure. Taken together, the paper is not a strong candidate for *eLife*, even if the authors can address all of the shortcomings in the experiments and interpretations mentioned above or below.

Other major criticisms:

The PICT approach requires Rapamycin, which is known to impair ribosome biogenesis. It's unclear from the genotypes provided whether both alleles of TOR1 in the diploid strains they used are the tor1-1 allele conferring resistance to Rap, which would seem to be required. Even if this is so, it seems important to show directly, if it's not in the literature already, that treatment of their PICT strain with rapamycin has no effect on Pol I recruitment or 35S pre-rRNA synthesis.

Figure 4: the very long time scale involved in the appearance or disappearance of the Pol I homodimers and Pol I:Rrn3 complexes raises the question of how important either one might be to regulating Pol I/Rrn3 occupancy of promoters, which might occur much more rapidly than either of these protein assembly reactions. It seems important to conduct ChIP experiments in the same regimen to determine how quickly promoter association is lost or regained in the transitions between growth and starvation – it might go to completion within a few minutes, which would be highly relevant to their interpretations.

It should be stipulated that the model in Figure 7 assumes that all of the Pol I is dimerized in starved cells, whereas the PICT assay only says that some detectable level of dimerization can be detected in starved cells without indicating the proportion of Pol I in this state. At this stage of their knowledge, it would be more prudent to depict a mixture of Pol I dimers and monomers present in starved cells with Pol I:Rrn3 complex assembly proceeding either from pre-existing monomers or following dissociation of dimers. This would make it easier to understand the multi-phase kinetics of Pol I:Rrn3 assembly and also take into account the fact that cycloheximide treatment of starved cells evokes even higher levels of dimer formation than starvation alone (implying that not all of the Pol I is dimerized in starved cells).

*Reviewer #2:*

The manuscript has two distinct parts, the first to study the in vivo formation of the inactive PolI dimer and its role in nutrient regulation of transcription in yeast, the second to present 4.9A Cryo-EM structures of yeast PolI and the PolI-Rrn3 complex. This reviewer does not feel competent to judge the validity or significance of these structures and so I will limit my review to the data on PolI dimer formation as a regulatory mechanism, Figure 1–Figure 4.

The authors use an adaption technique called PICT (Protein interactions from Imaging of Complexes after Translocation) that uses the ability of Rapamycin to mediate an interaction between FKBP and FRB. In this adaption of PICT, FKBP is fused to Tub4 and RFP, and FRB to the large subunit of PolI. Addition of rapamycin then causes FRB-PolI to be recruited to the RFP tagged spindle poles. The authors then study the co-localization of a GFP-PolI fusion with the RFP tag as a measure of PolI dimerization in rapid growth in rich medium and after nutrient withdrawal. Similarly, and Rrn3-GFP fusion is used to study the formation of the active PolI-Rrn3 complex. A rapamycin resistant yeast strain is used to avoid the known effects of this drug on the Tor pathway.

The technical approach is interesting and still novel. The question of whether or not an inactive PolI dimer forms in vivo and plays a part in regulation of the ribosomal RNA genes is an important one. This said, I found the data inadequate to support the key claims that nutrient deprivation or drug induced inhibition of protein synthesis or ribosome assembly leads to a significant accumulation of inactive PolI dimers and that could be reversed by refeeding or drug withdrawal.

Already from Figure 1 I found it very difficult convince myself that there was true co-localization of GFP and RFP and hence PolI dimer formation in the starved wt/wt situation. At best only a very small fraction of the GFP signal colocalizes with the RFP anchor, but a halo of GFP also forms around the anchor site (Figure 1). Why this halo forms is not discussed, and since no marker of the nuclear space is provided it is also very unclear what is being observed. The GFP halo around the RFP signal, very prominent in both wt/wt and A43ΔCt/A43ΔCt but present in most images throughout the manuscript, appears to make even a rough estimate of RFP-GFP co-localization near impossible. The quantitation protocol used (Materials and methods) makes no attempt to take this GFP halo into account as a (non co-localizing) background fluorescence. The problem is clearly demonstrated in Figure 1—figure supplement 1, where the background level of GFP even in these test examples is at least as large and often many times the co-localizing signal. Given this, not only is the quantification of the co-localizing PICT signal highly questionable, but it would be essential to demonstrate the degree to which the GFP signal is detected in the RFP optical channel, even a small overlap would give a very significant PICT signal.

The same problems occur in Figure 2, and again here the lack of definition of the nuclear volume and the varying distances between the anchor points suggest yet a further complication that each image represents a different cell cycle stage. The images shown in Figure 1 and Figure 2 do not convince me that it is possible to estimate in any reliable way the degree of co-localization of the anchor and PolI-GFP signals and in Figure 2 even the authors' quantitations for +- cycloheximide are well within the estimated errors. Also, the authors provide no quantitation of Diazaborine effects, nor a control to demonstrate that non-functional ribosomes actually form in their experiment as they claim. The title of this section "Non-functional ribosomes trigger Pol I homodimerization" is clearly not supported by the data.

For these reasons, it is my opinion that the data do not convincingly demonstrate PolI dimerization in vivo or that it is modulated on nutrient deprivation, transcription inactivation or inhibition of protein synthesis.

The evidence for an in vivo interaction between PolI and Rrn3 in Figure 3 is much more convincing. Here a co-localization is evident between the Rrn3-GFP and the anchored PolI-associated RFP signals. But the images do not convincingly show a reduction on nutrient starvation, and here again the GFP halo makes quantitation very uncertain. Further, the ChIP measurement of PolI and Rrn3 recruitment at the gene promoter reduces by over 80% in starved cells (Figure 3) and is not consistent with the only 50% reduction in PolI-Rrn3 estimated by PICT. (To be reliable the ChIP data should be extended to other amplicons both within and outside of the 35S transcribed region to show that the reduction in promoter signal is not due to chromatin accessibility changes after nutrient depletion, etc.) As is, the data provided suggest that the reduction in transcription on nutrient withdrawal is not due either to dissociation of the PolI-Rrn3 complex or the formation of inactive PolI dimers.

The time course data in Figure 4 is potentially interesting, but again is based on a techniques and quantitation regime that are wholly inadequate. Even if we accept that the data do in fact reflect in vivo changes, we have no idea what the relative signals mean in terms of the fraction of PolI in the different complexes. The PolI dimer signal might be rising on nutrient withdrawal and falling on refeeding, but we have no idea how much of PolI is undergoing this change. This is particularly clear from the refeeding data, where dimers appear to be rapidly eliminated, but PolI-Rrn3 complexes hardly increase over the same time period.

In short, the questions are interesting, the technique fascinating, but the data do not support the claims and interpretations.

*Reviewer #3:*

It's been known for quite a while that the Rrn3 transcription factor is needed for RNA Polymerase I activity. Several very recent papers, including this one, have shown that the structure of Rrn3 bound to Pol I appears mutually exclusive with dimerization of Pol I. This report goes beyond to the recent Nature Communications papers from Cramer and Schultz in providing in vivo evidence for this model of regulation. Using an improved fluorescent co-localization protocol (PICT), Pol I homodimers and Rrn3-Pol I heterodimers can be assayed under various conditions. Several treatments known to repress ribosomal transcription or to inhibit translation result in a shift from heterodimers to Pol I homodimers. Overall, I think this is an interesting and important paper that would be appropriate for *eLife*.

Specific comments:

For the PICT results, it's sometimes not clear how the individual cell panels illustrate the numbers shown in the bar graphs. For example, in Figure 1, why is the δ 43 heterozygote nearly WT on the bar graph, while the homozygote is zero? The pictures make it look like it should be the other way around. Please clarify exactly what the bar graphs are showing: simple overlap of peaks in two dimensions, or some 3D measurement? Perhaps more pictures would help.

In the last paragraph of the subsection “Dynamics in the assembly of Pol I complexes in response to nutrient availability”, the paper states that in the second stage after addition of nutrients, the pol I homodimers disappear, but with no increase in the Rrn3/pol I complex. Where do those polymerase molecules go, if not into the Rrn3 complex?

I like the idea in final paragraph of the Discussion, which draws parallels between the Rrn3-pol I and Mediator – pol II interactions. However, use of the word "holoenzyme" in the pol II system has been very messy and I would avoid it. To my mind, a holoenzyme should have all the activities needed for promoter recognition and transcription initiation. There were some early papers proposing a pre-assembled holoenzyme of pol II with all the general transcription factors, but those models haven't held up. Later papers started misusing holoenzyme to mean the Mediator – pol II complex. For similar reasons, I don't think Rrn3-pol I would qualify as a holoenzyme. I would avoid the word altogether so as not to distract from the concept that factor interactions with the stalk and surrounding regions might help bring RNA polymerases to the promoter.

[Editors' note: further revisions were requested prior to acceptance, as described below.]

Thank you for resubmitting your work entitled "The dynamic assembly of distinct RNA polymerase I complexes modulates rDNA transcription" for further consideration at *eLife*. Your revised article has been favorably evaluated by Kevin Struhl as the Senior Editor, Alan Hinnebusch as the Reviewing Editor and two reviewers.

The majority of the remaining issues have been raised by the Reviewing editor, as described below:

The authors have substantially improved the manuscript in several respects to address all of the major criticisms. They obtained a new mutation in the A43 CTT that specifically impairs homodimerization without affecting interaction with Rrn3, and showed that this mutation dampened the loss of Pol I:Rrn3 complexes from promoter DNA in starved cells, thereby providing evidence that dimerization is an important component of the down-regulation of Pol I recruitment and transcription. They better explained the partial cleft closure observed in their cryoEM structure of the Pol I monomer. They provided evidence that the A14 deletion does not simply destabilize Pol I as the means of impairing complex formation with Rrn3 and promoter binding of the complex. Importantly, they also added additional mutants that implicate helix 2 of the A14 subunit in Rrn3 association with Pol I, exploiting novel predictions of their Pol I:Rrn3 structure. They have also better explained the quantification of results from the PICT assay. As such, there are no remaining major criticisms that would require any additional experimentation to address.

However, there are two major issues with their description of the results.

First, their description of the new coIP experiment shown in Figure 1—figure supplement 4 is very confusing and incomplete. The only results that are straightforward are shown in lanes 1-2 and 5-6, which provide valuable evidence for Pol I homodimers in the soluble fractions obtained from cross-linked cells when the cells have been starved. In their "Responses to reviewers", they provide a somewhat more complete explanation of the last 3 lanes pertaining to the insoluble cell fractions, but even this doesn't make complete sense, and it was left out of the manuscript/legends completely. A coherent explanation of all of the lanes in Figure 1—figure supplement 4 is required.

Second, in the Discussion: the statement that "initial transcriptional inactivation is mainly driven by Pol I:Rrn3 disassembly." does not seem justified. The experiments in Figure 4 clearly show that the bulk of the dissociation of Pol I: Rrn3 from the promoter occurs within the first 20 min at a timepoint where Pol I:Rrn3 complexes are still very abundant and almost no Pol I homodimers have formed. It seems unavoidable that there is another regulatory mechanism apart from dissociation of Pol I:Rrn3 complexes and Pol I homodimerization responsible for the rapid and nearly complete loss of promoter-bound Pol I on starvation. From their statements at several places in the text, it appears that the authors actually agree with this view, but they don't state it directly in the Discussion and it is not depicted in the final model, which could be misleading to the field. The authors are urged to acknowledge more explicating in the Discussion and in their final model that there remains an important gap in knowledge about how the rapid dissociation of the Pol I:Rrn3 complex from the promoter is achieved on starvation.

*Reviewer #2:*

My major criticisms of the original manuscript concerned the use of the PICT technique and more precisely the analysis of images. The authors have tried to address these concerns, but it is clear that their quantification of Pol1 dimer and Pol1-Rrn3 complex formation is subject to considerable noise. This said, they now provide a detailed quantification protocol and have attempted to better explain the steps used to eliminate problems caused by the very high and variable GFP background signal. They have also improved their ChIP data, clearly validating their ChIP signal for Rrn3.

Given the novelty of the approach and the importance of better understanding Pol1 dimerization and whether or not it has a regulatory role, I feel that on balance the data should now be published.

*Reviewer #3:*

The revised version of this manuscript does a much better job explaining the PICT assay, and now that I understand how to look at the data I find that the authors have made a persuasive case. These experiments are very important in showing that yeast RNA pol I uses controlled dimerization, in competition with Rrn3 binding, to regulate transcription in response to nutrients and translational status. This work is nicely complemented by some structures that confirm and extend recent work from other groups. I am in favor of publication in *eLife*.

---

## [Author Response]

*Essential revisions:*

*The consensus of the reviewers is that the structural information in the second half of the paper is valuable, even if mostly confirmatory of other recently published structures, as are the structure-function studies, even if somewhat limited in scope. However, the structural insights into the transition from presumably inactive dimer to active monomer gleaned from the structures were either not dramatic or not well explained in the text/figures and should be highlighted better. In addition, the possible effects of the A14 subunit deletion on Pol I integrity and abundance would have to be examined.*

We thank the reviewers for the positive comments regarding the scientific value of the structural results. Following the Editor recommendation on the Pol I dimer to monomer structural description, we have included three major changes in our revised version: (i) text revision to more clearly highlight the structural transition, (ii) two additional panels in Figure 5 with zoom views of mobile domains and a schematic representation of the conformational change; and (iii) a new video (Rich Media File 2) to help visualize the conformational rearrangement. This is explained in detail in the response to reviewer 1. Regarding the effects of the A14 subunit deletion on Pol I levels, we performed western-blot analysis with antibodies against several subunits in the complex (A190, A135, A49, A43 and A34.5), and in all cases their levels are identical to those in the wild-type strain.

*Regarding the results obtained using the PICT assay in the first part of the paper, there are important concerns about whether the results are sufficient to conclude, as the title of the paper implies, that formation/disassembly of the homodimer versus Pol I-Rrn3 complexes are important events regulating rDNA transcription in response to nutrient availability. All three reviewers had difficulty understanding how the PICT results have been quantified; Rev. #2 had the most serious concerns, and has explained the difficulties thoroughly in his/her review. The large halo of background GFP signal compared to the small amount associated with the tethered RFP makes quantification of the co-localizing PICT signal highly questionable; and the images in Figure 1 and Figure 2 raise doubts about whether the degree of co-localization of the anchor and Pol I-GFP signals can be reliably measured. The PICT homodimerization data seem to indicate that only a small fraction of the Pol I exists in homodimers, and the Pol-Rrn3 PICT data indicate only a modest reduction in this complex (which might merely reflect reduced Rrn3 abundance) despite a strong loss of Rrn3/Pol I occupancy of the RDN promoter measured by ChIP. The analysis of the kinetics of forming/dissociating these complexes by PICT do not indicate tight coordination of Pol-Rrn3 dissociation and homodimer formation, and provide no evidence that homodimerization is driving dissociation of Pol I-Rrn3 complexes. The rates of forming/dissociating these complexes measured by PICT are quite slow and have not been compared to the rates of Rrn3/Pol I association/dissociation from the promoter measured by ChIP. As such, formation/dissociation of these protein complexes may be secondary to the key events that regulate promoter occupancy at much higher rates. In addition, the conclusions that inactive ribosomes trigger Pol I dimerization are not substantiated by the data and represent an overinterpretation of the data. Thus, while the PICT assay might be providing evidence for Pol I dimerization in cells that is triggered by nutrient starvation, provided the criticisms raised by Rev. #2 can be satisfied, it is unclear that dimerization is extensive enough and occurs with the proper kinetics to represent an important mechanism for down-regulating Pol I function, as opposed to a secondary event that might simply protect idle Pol I molecules from degradation. One way to support your claim would be to construct a mutation in the A43-CTT that specifically impairs homodimerization and show that this mutation abolishes homodimerization by PICT and also dampens loss of Pol 1:Rrn3 promoter complexes in starved cells measured by ChIP.*

We would like to thank all reviewers for their comments on the PICT assay. We apologize for the lack of information on how the images from PICT assays were analyzed, and for missing figures illustrating this. We have now detailed the procedure in the ‘Materials and methods’ section and also included a clear definition of the recruitment score used to quantify PICT (see comments to reviewers 1 and 2 for a deeper explanation). In addition, we appended a new figure supplement (Figure 1—figure supplement 2) with a step-by-step illustration of the image analysis to detect Pol I homodimerization in starving cells. We agree that quantifying the recruitment to the anchoring platform is not trivial and that the GFP halo around the anchor further complicates this quantification. Nonetheless, our image analysis protocol allows for specific segmentation of the GFP signal recruited to the anchoring platforms, regardless of the background. We agree that images used to illustrate the PICT assays through the manuscript conveyed the false impression that the intensity of the GFP halo varies among the different conditions. The negligible effect of the GFP halo in the detection of Pol I complexes and computation of the recruitment score is extensively discussed in the comments to reviewer 2.

Regarding the influence of complex formation and disruption on rDNA transcriptional regulation, we have performed new experiments that shed light on our interpretation. To support our claim that homodimerization influences Pol I transcription, we designed a new mutant (A43ΔCt, Δ307-326) that specifically abolishes Pol I ability to homodimerize (Figure 1 and Figure 3). Upon starvation, this strain presents higher levels of Pol I and Rrn3 associated to rDNA (Figure 3). Nonetheless, these levels remain lower than in growing cells, indicating that another mechanism also contributes to inactivate Pol I transcription (see specific response to reviewer 1). In addition, we measured promoter association of A190 and Rrn3 along two hours after cells were deprived from nutrients (Figure 4). Our results show that full promoter dissociation occurs with comparable dynamics as Pol I–Rrn3 disassembly, i.e. within 2 hours from starvation. Moreover, we studied the behavior of a chimeric strain where Rrn3 and A43 are covalently fused (Laferte et al., 2006), so that no free monomeric or dimeric Pol I can form. ChIP data show higher association levels of A190 than the wild-type along the entire rDNA gene (Figure 3—figure supplement 2), indicating that, in the absence of Pol I dimers and increased Pol I–Rrn3 levels, rDNA transcription is upregulated in starved cells. The fact that we detect Pol I–Rrn3 complexes in the wild-type strain when promoter association is null (Figure 3 and Figure 4), suggests that Pol I–Rrn3 levels alone are not enough to drive transcription inactivation in starved cells. A TA-loop mutant that specifically increases Pol I–Rrn3 levels without affecting promoter association in starving conditions further supports this. As this mutant also presents unperturbed levels of Pol I dimers, it confirms the observation from time-course PICT that assembly/disassembly of Pol I–Rrn3 does not drive Pol I dimerization. Finally, we have also more accurately interpreted our data on Pol I dimerization upon drug treatment, as explained in detail for reviewer 1. To account for all these results, we modified the main text at different sections (see specific sections in responses to reviewer 1). Moreover, to better express the conclusions of our work, we modified the manuscript title, as also suggested by the Editor (see below).

*Specific recommendations:*

*1) It would be very helpful if they had an independent assay(s) for dimerization. Two choices. First, they could formaldehyde crosslink under the various in vivo situation and do a co-IP with essentially the same constructs they already have. Second, they could replace the targeting tags with Gal4, which would allow Gal4-Pol I fusions to targeted to Gal4 binding sites (assayed by ChIP). If the model is correct, one should see the non-Gal4 version of Pol I associating with Gal4 binding sites under starvation. Both of these approaches might be better for kinetics and quantification, but the real virtue of them is that they are independent assays for dimerization.*

We thank the Editor for this useful recommendation. We have performed co-IP experiments after crosslinking on a diploid strain harboring TAP and MYC-tags at the C-termini of each A190 allele. After clarification of whole cell extracts to separate the soluble and chromatin insoluble fractions, we used both fractions to pull down A190-TAP with IgG resin. Thereafter, pulled down proteins were de-crosslinked at 65 ºC for several hours and we analyzed co-IPed A190-MYC by western blot. As shown in Figure 1—figure supplement 4, we could detect Pol I homodimers in the soluble fraction of starved cells, while no dimer could be observed in the same fraction of growing cells. On the opposite, and as expected, we detect A190-MYC in the chromatin fraction in growing conditions, most likely corresponding to tandemly elongating polymerases. We detect tiny amounts of Pol I dimers in the chromatin fraction of starving cells, probably as a result of some protein contamination from the soluble fraction, as suggested by the presence of low levels of Pgk1 in this fraction (panel B), or due to minor amounts of Pol I still attached to DNA after two hours of starvation. DNAse I treatment of the precipitated A190-TAP confirms that detected A190-MYC in starved cells was bound to the DNA. These results have been included in the revised manuscript, as follows:

“In addition, we performed co-immunoprecipitation experiments after crosslinking, using a diploid strain where one A190 allele was tagged with TAP (A190-TAP) and the second with MYC (A190-MYC). Homodimerization could be observed in the soluble fraction of starved cells only (Figure 1—figure supplement 4). Overall, these results indicate that cells induce the formation of Pol I homodimers specifically upon starvation.”

*2) The authors haven't demonstrated "regulation", so their title needs to be softened. What they have "shown" is that the dimerization and change in complexes is associated with regulation.*

Following the Editor recommendation and in the view of the new experiments, we have now changed the title to “The dynamic assembly of distinct RNA polymerase I complexes modulates rDNA transcription”, which more accurately reflects the significance of our results.

*Reviewer #1:*

*[…] General critique:*

*Previous work on Pol I in yeast has shown that Rrn3 binding to the enzyme is required for Pol I recruitment to the promoter, and that Pol I can form homodimers in solution and was crystallized as a homodimer with structural features indicating an inactive conformation. This paper uses the PICT assay to provide convincing evidence that Pol I homodimers can form in cells, but only in stress conditions including starvation for carbon and nitrogen, arrest of translation elongation by cycloheximide, or treatment with a drug shown to impair ribosome biogenesis. The evidence that homodimerization occurs in cells is firm, and the fact that it was observed only in starved/stressed cells supports its role as a regulatory mechanism for inactivation of Pol I under conditions where new ribosome synthesis is unwanted. Unfortunately, however, there is no direct evidence establishing its importance in down-regulating Pol I function. In fact, the kinetics of homodimerization versus PolI:Rrn3 interaction shown in Figure 4 reveal that the initial, and most rapid phase of dissociation of Pol I:Rrn3 on the shift from growth to starvation precedes any appreciable formation of Pol I dimers, and hence, is clearly not being driven, at least initially, by dimerization. What is needed in my view is a mutation that specifically impairs homodimerization, such as in the A43-CTT, and to show that this mutation would dampen the loss of Pol 1:Rrn3 complexes from promoter DNA in starved cells, thereby showing that dimerization is an important component of the down-regulation of Pol I recruitment/transcription. Unfortunately, the mutation they examined that simply deletes the entire A43-CT has the complication of also, inexplicably, impairing Pol I:Rrn3 interaction and promoter recruitment. Thus, a more surgical mutational approach would be needed to achieve this important goal.*

We thank the reviewer for this useful suggestion. In spite of the challenge, we succeeded in finding a mutant that abolishes homodimerization while maintaining Pol I–Rrn3 levels (A43ΔCt, Δ307-326). When this strain is cultured in starving medium, ChIP experiments show that the promoter levels of Pol I and Rrn3 are higher than in the WT strain. This proves that homodimerization is required for full down-regulation of Pol I recruitment/transcription and improves the significance of our results. To improve simplicity for the reader, we eliminated the former A43ΔCt mutant (Δ277-326) and used the same nomenclature for our more surgical mutant, so that in the revised manuscript A43ΔCt corresponds to A43Δ307-326. The results of this mutant appear at two sections in the revised manuscript:

“Therefore, we studied Pol I dimerization upon partial deletion of this structural element (A43ΔCt, Δ307-326). In this mutant, Pol I homodimerization is impaired (Figure 1), confirming the observation derived from structural data.”

“To further investigate the effect of starvation on Pol I transcription, we used our A43 C-terminal truncation abolishing Pol I dimerization. […] Furthermore, in these conditions, association of Pol I along the rDNA gene and of Rrn3 at the promoter increase about 6-fold with respect to the wild-type.”

In addition, we studied the behavior of a chimeric strain where A43 is covalently fused to Rrn3 (Laferte et al., 2006), so that all Pol I is complexed to Rrn3 and no free monomeric or dimeric Pol I can form. This strain presents higher association levels of A190 along the entire rDNA gene and recovers faster from starvation than the wild-type. This indicates that, when Pol I dimers cannot form and Pol I–Rrn3 levels are increased, rDNA transcription is upregulated in starving cells. The corresponding manuscript section now reads:

“We also used a strain expressing a Pol I–Rrn3 chimera that cannot form Pol I homodimers (Laferte et al., 2006). […] Moreover, recovery of CARA from nutrient-depleted medium is faster than for wild-type cells (Figure 3—figure supplement 2).”

*The PICT assay also gives evidence of reduced Pol I:Rrn3 association in starved cells by ~50%, and there is an even greater reduction in Pol I/Rrn3 occupancy of the Pol I promoter measured by ChIP. However, the Western in Figure 1—figure supplement 2 shows reduced Rrn3 abundance that could be as much as 50%, which would lead to a different and less interesting interpretation of the PICT data, with no evidence for a weaker Pol I:Rrn3 association occurring in starved cells.*

We agree that the PICT technique does not provide evidences about the affinity of the interaction between Pol I and Rrn3, but only about relative levels of the complex. As the reviewer pointed out, the levels of Rrn3 drop upon starvation. This suggests that degradation of Rrn3 might be a mechanism that the cell employs to regulate Pol I–Rrn3 levels during the first stages of starvation. We performed western-blots of Rrn3 and A190 at different time points (Figure 4—figure supplement 1). Although upon starvation the levels of Rrn3 roughly correlate with those of Pol I–Rrn3 as detected by PICT, in the reverse transition the levels of Rrn3 recover steadily while Pol I–Rrn3 levels follow a different dynamic that partially correlates with Pol I homodimer disassembly. Moreover, the A14△TAloop mutant presents higher levels of Pol I–Rrn3 while western-blot shows that Rrn3 levels are unaltered respect to the wild-type. Therefore, Rrn3 degradation might be an important mechanism to regulate Pol I–Rrn3 levels during the first stages upon starvation, but not in cells starved for longer times. Similarly, upon re-feeding, synthesis of fresh Rrn3 likely plays an important role to control Pol I–Rrn3 levels, but this is not sufficient to explain the dynamics observed, which suggest that Pol I homodimers disassembly might also contribute to activate transcription, as supported by the novel A43ΔCt mutant described above. In summary, we conclude that Pol I–Rrn3 levels are an important regulatory mechanism of Pol I activation in growing cells and during the first minutes upon starvation, but in cells starved for longer times they are not sufficient to determine rDNA transcription. This is in agreement with previous observations of rDNA inactivation by inhibition of TOR signaling (Philippi et al., 2010). The revised manuscript now reads:

“We show that nutrient depletion induces a rapid reduction in the levels of Pol I–Rrn3, which correlates in time with a marked decrease in promoter association of both Pol I and Rrn3. […] This is in agreement with previous observations of rDNA transcription inactivation by inhibition of TOR signalling (Philippi et al., 2010).”

*The cryo-EM analysis of the Pol I monomer at 4.9 Å resolution and the Pol I::Rrn3 complex at 7.7 Å resolution are valuable, although another group recently published a Pol I-Rrn3 complex at higher resolution (4.9 Å) that seems to contain all of the same key features and allows the same insights into this important intermediate in Pol I activation as those described here-thus diminishing the significance of the current structure of the complex. The higher resolution of their monomeric Pol I compared to that recently published by another group, should allow a superior comparison between the monomeric and dimer Pol I (from the crystal structure), from which they conclude that the transition from dimer to monomer "involves partial cleft closure and increased flexibility of critical motifs". However, the relevant figure (Figure 5) does not make a convincing case for the cleft closure in the monomer, and the accompanying text (subsection “Cryo-EM structures of monomeric Pol I alone and in complex with Rrn3”) did not help much to convince me. In addition, it seems possible that the critical motifs are flexible in both the monomer and dimer but were fixed in the dimer by crystal contacts.*

Partial cleft closure in the dimer to monomer transition is half the closure required to achieve the transcription-competent state and, importantly, no further closure is observed upon Rrn3 binding. Both observations highlight the relevance of this conformational change. In order to improve clarity in our text, we have included a more detailed explanation of this structural transition that involves two events: (i) increased flexibility of 3 domains, the stalk, the DNA-mimicking loop and the A12.2 C-terminal Zn-ribbon; and (ii) partial cleft closure by approach of the clamp and protrusion domains. The revised manuscript incorporates these changes in the subsection “Cryo-EM structures of monomeric Pol I alone and in complex with Rrn3”. As a second major change, we have included new panels in Figure 5 that present close-up views of the domains undergoing the structural transitions described in the text, as well as a schematic representation of the conformational changes (Figure 5). To further improve the graphical presentation, we decided to split the resulting figure into two main figures, one describing the transitions from dimeric to monomeric Pol I (Figure 5) and the other showing the transitions from monomeric to Rrn3-bound Pol I (Figure 6). As a third major change, we included an additional Rich Media File 2 showing a morph between the dimeric and monomeric states of free Pol I.

Regarding crystal contacts, domains that become flexible upon monomerization (stalk, DNA mimicking loop and A12.2 C-terminal Zn-ribbon) are not directly involved in crystal maintenance. Moreover, the cryo-EM structure of Pol I dimers in solution at 7.5 Å resolution is virtually identical to that of the crystals (Pilsl et al., 2016). Therefore, it is not possible that these critical motifs are fixed in the dimer structure as a result of the crystal contacts. A sentence stating this has been included in our revised manuscript:

“When compared with the crystal structure of dimeric Pol I (Engel et al., 2013; Fernandez-Tornero et al., 2013), which is essentially identical to dimeric Pol I in solution (Pilsl et al., 2016), our monomeric Pol I structure presents a rearranged cleft entrance where the clamp coiled-coil and the protrusion approach by about 4 Å (Figure 5).”

*The experiments showing that deleting the A14 subunit weakens Pol I:Rrn3 interaction by PICT assay and reduces Pol I/Rrrn3 occupancy at the promoter by ChIP assay are valuable, but they have not ruled out the possibility that removing this subunit perturbs other Pol I subunits, e.g. A43 with which it seems to interact with extensively, and thus impairs Pol I interaction with Rrn3 indirectly by disrupting A43:Rrn3 contacts rather than eliminating the A14:Rrn3 interactions evident in the cryoEM structure. It is even possible that the steady state level of Pol I is reduced by the A14 deletion and contributes to the reduced yield of Pol I:Rrn3 interactions and Pol I recruitment. Thus, the effects of the A14 subunit deletion on Pol I integrity and abundance should have been examined. But even more importantly, the authors should have used the molecular details of the cryoEM structure to more surgically disrupt the A14:Rrn3 interaction by truncating or substituting helix 2 of A14 in a way designed not to disrupt A14 interactions with other Pol I subunits. This seems like a missed opportunity to exploit their structure.*

To rule out the possibility that mutations of A14 perturb other Pol I subunits, we performed western-blot analysis against several subunits in the complex (Figure 7—figure supplement 1). For the complete deletion of A14 we tested subunits A190, A135, A49, A43 and A34.5. For the other mutants we tested A190. For all the mutants, we also evaluated the Rrn3 levels. In all cases the levels of detected proteins were similar to those in the wild-type strain. To further exploit our structural data, we designed two additional mutants in subunit A14. Following the reviewer’s suggestion, we have produced a deletion mutant that includes helix-α2 and the A14-Ct tail, the latter being innocuous for Rrn3 binding. Additionally, we also designed a surgical point mutant within helix-α2, R91E. Cells with helix-α2 mutated (deletion or point mutation) present lower levels of Pol I–Rrn3 and lower occupancy at the rDNA promoter of Pol I and Rrn3, features that resemble △A14 cells. Together with the A14ΔTAloop mutant (see below), these results confirm that the activation of Pol I in growing cells is regulated through the levels of Pol I–Rrn3 assembly. This is presented in the revised manuscript (subsection “The stalk subunit A14 influences rDNA promoter association”, second paragraph) and accompanying figure (Figure 7), as well as in the Discussion (subsection “The Pol I stalk as a sensing platform of the cell state”).

*In summary, the paper represents a collection of three different lines of work joined by the theme of regulating Pol I:Rrn3 interaction. The evidence for homodimer formation in cells is interesting and valuable but it is unclear that it constitutes an important means of impeding Pol I:Rrn3 association and promoter recruitment rather than being a byproduct of the loss of this complex by other means that serves only to protect Pol I from degradation. The cryoEM structures are valuable, but a Pol I:Rrn3 complex of higher resolution was already published. The structure-function analysis of A43 and A14 did not exploit the cryoEM structure, as it involved deletions of large segments that are not present at their interfaces with Rrn3. The result that deleting the TA loop in A14 leads to greater Pol I:Rrn3 association and promoter occupancy is the most interesting finding, but the molecular mechanism is not obvious from the structure. Taken together, the paper is not a strong candidate for eLife, even if the authors can address all of the shortcomings in the experiments and interpretations mentioned above or below.*

We believe that the three lines of work and the diversity of methods that we employed to study the influence of Pol I complexes on transcriptional activation represent a major strength of our manuscript. Overall, the manuscript has been improved in different ways:

1) Pol I homodimerization is now confirmed by Co-IP after crosslinking experiments (Figure 1—figure supplement 4). Moreover, to evaluate if Pol I homodimerization is just a byproduct of Pol I–Rrn3 complex clearance, we analyzed the levels of Pol I homodimers in cells with perturbed levels of the Pol I–Rrn3 complex. Using the A14ΔTAloop, which exhibits a 2-fold increment in Pol I–Rrn3 levels in starving conditions, we could not detect any change in Pol I dimerization compared to the WT (Figure 7—figure supplement 2). This observation suggests that Pol I homodimerization is not a consequence of Pol I-Rrn3 clearance, but a process that is specifically induced upon starvation.

2) New mutants allow specific targeting of Pol I complexes to discern the contribution of each assembly in Pol I regulation. As mentioned, we have further exploited our Pol I–Rrn3 structure to design mutants that show the biological relevance of subunit A14. Helix-α2 mutants specifically disrupt Pol I–Rrn3 assembly and show that Pol I–Rrn3 complexes are required for proper Pol I activation in growing cells. Moreover, cells expressing a mutated form of A14 lacking the TA-loop, show higher levels of Pol I–Rrn3, both in growing and starved cells (Figure 7 and Figure 7—figure supplement 2). Since Rrn3 and A190 levels are equivalent to WT cells (Figure 7—figure supplement 1), this confirms the role of the TA-loop in regulating Pol I–Rrn3 levels independently of total Rrn3 levels. In addition, ChIP experiments show that in growing cells higher levels of Pol I–Rrn3 are enough to strength the activation of Pol I. However, under starvation higher Pol I–Rrn3 assembly is not enough to increase the recruitment of Pol I to the promoter. These results indicate that levels of Pol I–Rrn3 regulate Pol I activation in growing cells but are not sufficient for inactivation after long nutrient depletion. Interestingly, our more surgical A43ΔCt mutant (Δ307-326), which specifically impairs Pol I dimerization, dampens loss of Pol I and Rrn3 at the promoter (Figure 1 and Figure 3). This shows that Pol I homodimerization in starved cells contributes to down-regulate Pol I–Rrn3 levels and subsequent recruitment to rDNA. Thus, we propose that the cell uses homodimerization to down regulate Pol I in starved cells.

3) We used ChIP to study the dynamics of Pol I activation upon perturbation in nutrient availability (further described below). Down regulation of Pol I upon nutrient deprivation follows a similar pattern than Pol I–Rrn3 disassembly. This is in agreement with the mutagenic analysis and suggests that the levels of Pol I–Rrn3 dictate activation of Pol I transcription in growing cells and the first minutes upon nutrient deprivation. Upon re-feeding, rDNA association follows a linear behavior, different to the dynamics of Pol I–Rrn3 assembly. We hypothesize that this reflects the combination of Pol I release from homodimers and the increment in the levels of Pol I–Rrn3.

In conclusion, we believe that our manuscript represents a significant advance in the field.

*Other major criticisms:*

*The PICT approach requires Rapamycin, which is known to impair ribosome biogenesis. It's unclear from the genotypes provided whether both alleles of TOR1 in the diploid strains they used are the tor1-1 allele conferring resistance to Rap, which would seem to be required. Even if this is so, it seems important to show directly, if it's not in the literature already, that treatment of their PICT strain with rapamycin has no effect on Pol I recruitment or 35S pre-rRNA synthesis.*

We have clarified the genotype in [Supplementary-material SD1-data]. All diploid strains used had the two *tor1-1* alleles as well as the two *fpr1* alleles deleted. Following the reviewer’s suggestion, we performed ChIP analysis in our tor1-1 mutant strain and concluded that Rapamycin treatment has no effect on Pol I association to the rDNA promoter. The corresponding ChIP experiment has been included as a new panel A in Figure 1—figure supplement 3. The revised manuscript now reads:

“Since Pol I transcription is down-regulated by rapamycin, all subsequent experiments were performed in rapamycin-insensitive strains carrying the tor1-1 mutation (Helliwell et al., 1994), so that addition of this compound has no effect on Pol I association to rDNA promoters (Figure 1—figure supplement 3).”

*Figure 4: the very long time scale involved in the appearance or disappearance of the Pol I homodimers and Pol I:Rrn3 complexes raises the question of how important either one might be to regulating Pol I/Rrn3 occupancy of promoters, which might occur much more rapidly than either of these protein assembly reactions. It seems important to conduct ChIP experiments in the same regimen to determine how quickly promoter association is lost or regained in the transitions between growth and starvation – it might go to completion within a few minutes, which would be highly relevant to their interpretations.*

We conducted ChIP experiments at similar time intervals as PICT after nutrients were severely altered. Because performing ChIP experiments at many time points is very challenging, we chose time points that cover the different stages observed in complex assembly by PICT. Our results show that rDNA promoter association of Pol I and Rrn3 extinguishes and recovers with kinetics in the same time scale as changes observed in the levels of Pol I homodimers and Pol I–Rrn3 complexes detected by PICT. Upon nutrient deprivation, promoter association drops exponentially and correlates with changes in Pol I–Rrn3 levels. The rate curve resembles that of Pol I–Rrn3 clearance, but the latter only reaches 40% of the initial value, in accordance with the overall levels of Rrn3 as shown by western-blot. This is in agreement with the experiments of A14 mutants and the Rrn3-A43 fusion strain (see above). Altogether, the data show that Pol I–Rrn3 levels influence Pol I activation in growing cells and during the first minutes upon starvation, but they are not sufficient to regulate Pol I inactivation in long-term starved cells. Upon nutrient re-supplementation, promoter association increases steadily while Pol I–Rrn3 formation follows different dynamics, where Pol I–Rrn3 levels grow at an early stage, remain constant during a second stage and then increase again up to growing levels. This suggests that Pol I homodimers represent a rapid enzyme supply to generate Pol I–Rrn3 complexes. This is supported by the observation that cells where Pol I homodimerization is impaired (new A43ΔCt mutant) present higher levels of Pol I and Rrn3 at the promoter and of the former within the gene. This mutant also suggests that in cells starved for 2 hours, Pol I homodimerization accounts for about 60% of Pol I inactivation, according to Pol I promoter. The description of the new ChIP experiments has been included in the revised manuscript, which now reads:

“ChIP experiments performed at an equivalent regime show that promoter association of both A190 and Rrn3 drops by two thirds within the first 10 minutes from starvation but requires about 2 hours to reach completion (Figure 4, left). […] Overall, these results indicate that cells respond to nutrient availability by differentially adjusting the levels of Pol I homodimers and Pol I–Rrn3.”

*It should be stipulated that the model in Figure 7 assumes that all of the Pol I is dimerized in starved cells, whereas the PICT assay only says that some detectable level of dimerization can be detected in starved cells without indicating the proportion of Pol I in this state. At this stage of their knowledge, it would be more prudent to depict a mixture of Pol I dimers and monomers present in starved cells with Pol I:Rrn3 complex assembly proceeding either from pre-existing monomers or following dissociation of dimers. This would make it easier to understand the multi-phase kinetics of Pol I:Rrn3 assembly and also take into account the fact that cycloheximide treatment of starved cells evokes even higher levels of dimer formation than starvation alone (implying that not all of the Pol I is dimerized in starved cells).*

We appreciate the reviewer’s comment but we did not intend to suggest that all of the Pol I is dimerized in starved cells. We have changed former Figure 7, now Figure 8, to better integrate the conclusions and hypothesis derived from our results. The revised manuscript now reads:

“Therefore, both Pol I–Rrn3 and Pol I dimerization modulate rDNA transcription, which allows us to propose a model (Figure 8). […] Upon refeeding from starvation, we propose that available Pol I–Rrn3 complexes are rapidly recruited for transcription, while disruption of Pol I homodimers provides fresh Pol I that can interact with Rrn3 to increase Pol I–Rrn3 complexes and further activate rDNA transcription.”

*Reviewer #2:*

*The manuscript has two distinct parts, the first to study the in vivo formation of the inactive PolI dimer and its role in nutrient regulation of transcription in yeast, the second to present 4.9A Cryo-EM structures of yeast PolI and the PolI-Rrn3 complex. This reviewer does not feel competent to judge the validity or significance of these structures and so I will limit my review to the data on PolI dimer formation as a regulatory mechanism, Figure 1–Figure 4.*

*The authors use an adaption technique called PICT (Protein interactions from Imaging of Complexes after Translocation) that uses the ability of Rapamycin to mediate an interaction between FKBP and FRB. In this adaption of PICT, FKBP is fused to Tub4 and RFP, and FRB to the large subunit of PolI. Addition of rapamycin then causes FRB-PolI to be recruited to the RFP tagged spindle poles. The authors then study the co-localization of a GFP-PolI fusion with the RFP tag as a measure of PolI dimerization in rapid growth in rich medium and after nutrient withdrawal. Similarly, and Rrn3-GFP fusion is used to study the formation of the active PolI-Rrn3 complex. A rapamycin resistant yeast strain is used to avoid the known effects of this drug on the Tor pathway.*

*The technical approach is interesting and still novel. The question of whether or not an inactive PolI dimer forms in vivo and plays a part in regulation of the ribosomal RNA genes is an important one. This said, I found the data inadequate to support the key claims that nutrient deprivation or drug induced inhibition of protein synthesis or ribosome assembly leads to a significant accumulation of inactive PolI dimers and that could be reversed by refeeding or drug withdrawal.*

*Already from Figure 1 I found it very difficult convince myself that there was true co-localization of GFP and RFP and hence PolI dimer formation in the starved wt/wt situation. At best only a very small fraction of the GFP signal colocalizes with the RFP anchor, but a halo of GFP also forms around the anchor site (Figure 1). Why this halo forms is not discussed, and since no marker of the nuclear space is provided it is also very unclear what is being observed. The GFP halo around the RFP signal, very prominent in both wt/wt and A43ΔCt/A43ΔCt but present in most images throughout the manuscript, appears to make even a rough estimate of RFP-GFP co-localization near impossible. The quantitation protocol used (Materials and methods) makes no attempt to take this GFP halo into account as a (non co-localizing) background fluorescence. The problem is clearly demonstrated in Figure 1—figure supplement 1, where the background level of GFP even in these test examples is at least as large and often many times the co-localizing signal. Given this, not only is the quantification of the co-localizing PICT signal highly questionable, but it would be essential to demonstrate the degree to which the GFP signal is detected in the RFP optical channel, even a small overlap would give a very significant PICT signal.*

This reviewer mentions a list of points that are key to correctly interpret the PICT assay. We agree that the Methods section and some of the figures lack clarity and might have been misleading. For this reason we appreciate his/her comments and have modified the manuscript to address them as follows:

The vast majority of A190 accumulates in a sub-compartment of the nucleus. We imaged the perimeter of the nucleus using the Nic96-RFP marker (a component of the nuclear pore complex) in cells with the genetic background used for PICT (i.e. *tor1-1* mutant and deleted fpr1) and that express A190-FRB-GFP (OGY0346). These cells showed that some A190-FRB-GFP signal is also observed through the entire nucleus, which the reviewer called GFP “halo”. Since the anchoring platform (Tub4-RFP-FKBP) resides in the nuclear envelope, depending on the orientation from where each cell is imaged, the RFP spot corresponding to the anchor might be surrounded by more or less of this GFP halo. Since in the in focus field of view the nucleus of each cell is oriented randomly, we expect a high variability. As shown in Figure 1—figure supplement 3, growing and starving cells have comparable mean GFP intensity in the area occupied by the anchor. Nevertheless, in both conditions, cell-to-cell variability is remarkable, with a standard deviation higher than 30% of the mean. In summary, the GFP halo does not form as a result of the rapamycin or the media used, but probably reflects the localization of a small population of A190 in yeast cells. We have corrected all PICT figures in the manuscript to include cells that present a GFP halo signal that is representative of each sample. A deeper study of the A190 diffused in the nucleus goes beyond the scope of this manuscript.

Additionally, we quantified the impact of the incubation media in the recruitment score by measuring the recruitment of A190 fused to FRB and GFP (A190-FRB-GFP). As expected, there was no significant difference between the recruitment score determined in growing and starved cells.

Our image analysis workflow, which is now fully detailed in Figure 1—figure supplement 2, has been designed to guarantee a specific detection of spots in the GFP channel, even in the presence of strong but smoothly varying background signal. This is achieved by local thresholding adapted to expected spots geometry, and the validation of the detected connected particles. Furthermore, only residual false positives GFP spots overlapping with a detected anchor actually count toward the recruitment score. Since anchors are sparsely located, and are virtually detected flawlessly, this only happens very marginally (see Figure 1—figure supplement 2). Overall, recruitment scores are highly reproducible across experiment conditions and exhibit a low standard deviation.

As mentioned, the mean GFP halo intensity is comparable in growing and starved cells. Since the quantification of the recruitment score is done on multitude of cells (see comment to reviewer 1), the cell-to-cell variability is averaged out. For this reason, it is not necessary to take into account this GFP signal in order to establish relative differences between our samples.

Figure 1—figure supplement 1 shows the intensity profile in the GFP (prey-GFP) and the RFP (Tub4-RFP-FKBP) channel. The profile of the GFP and RFP channels are shown in arbitrary units so that they do not overlap and the brightest pixel of each channel can be identified easily. In the three examples, the GFP signal that co-localizes with Tub4-RFP-FKBP is higher than the surrounding GFP signal.

The optical system of our microscope ensures that bleed-through from the GFP to the RFP channel is negligible since we have used specific combination of optimized beam splitters, excitation and emission filters (cubes) for each fluorophore type and since the images were recorded sequentially. As mentioned earlier, the vast majority of the A190-GFP signal is in the nucleolus. If there was bleed-through from the GFP to the RFP channel, we should see this reflected in an apparent RFP signal localizing in this sub-compartment of the nucleus. Among other controls, the blank experiments (Figure 1—figure supplement 3) show that there is no detectable GFP signal in the RFP channel or RFP signal in the GFP channel, as there is no colocalization between the two fluorophores before adding rapamycin. We have detailed the optics of our microscope in the ‘Materials and methods’ section.

In summary, detection of Pol I homodimers by PICT cannot result from artifacts caused by the GFP halo, the incubation in different medium or bleed-through between the GFP and the RFP channels.

*The same problems occur in Figure 2, and again here the lack of definition of the nuclear volume and the varying distances between the anchor points suggest yet a further complication that each image represents a different cell cycle stage. The images shown in Figure 1 and Figure 2 do not convince me that it is possible to estimate in any reliable way the degree of co-localization of the anchor and PolI-GFP signals and in Figure 2 even the authors' quantitations for +- cycloheximide are well within the estimated errors. Also, the authors provide no quantitation of Diazaborine effects, nor a control to demonstrate that non-functional ribosomes actually form in their experiment as they claim. The title of this section "Non-functional ribosomes trigger Pol I homodimerization" is clearly not supported by the data.*

*For these reasons, it is my opinion that the data do not convincingly demonstrate PolI dimerization* in vivo *or that it is modulated on nutrient deprivation, transcription inactivation or inhibition of protein synthesis.*

We have engineered anchoring platforms at the spindle pole body (SPB). During the cell cycle, SPB duplicates and separates to the daughter and mother cell. Therefore, cells hold 1 or 2 anchoring platforms depending on the cell cycle stage they are at during imaging. Starved cells also hold 1 or 2 anchoring platforms. When holding 2 anchoring platforms, the observed anchor-to-anchor distance spread is large and depends on the cell cycle stage at the moment of nutrient deprivation. These differences, however, are averaged out in our quantifications because the overall PICT recruitment score is computed over a minimum of 67 cells for each biological replicate (3 biological replicates). We apologize because some of the images we had chosen presented highly varying anchor to anchor distances, suggesting that samples had been analyzed at different cell cycle stages. We have now chosen cell images to correct this misunderstanding. The new Figure 1—figure supplement 2 shows the detailed workflow of the PICT image analysis and a field of view with cells (with different number of anchoring platforms and anchor to anchor distances).

Following the reviewer’s suggestion, we have performed a quantitative PICT assay to estimate the recruitment score of Pol I homodimers upon diazaborine and cycloheximide treatment in growing cells (Figure 2). As mentioned in previous comments, we have improved the clarity of the figures for PICT experiments by choosing representative cells for the GFP halo intensity and cell cycle stages, as well as by adjusting the contrast of the GFP image.

*The evidence for an in vivo interaction between PolI and Rrn3 in Figure 3 is much more convincing. Here a co-localization is evident between the Rrn3-GFP and the anchored PolI-associated RFP signals. But the images do not convincingly show a reduction on nutrient starvation, and here again the GFP halo makes quantitation very uncertain. Further, the ChIP measurement of PolI and Rrn3 recruitment at the gene promoter reduces by over 80% in starved cells (Figure 3) and is not consistent with the only 50% reduction in PolI-Rrn3 estimated by PICT. (To be reliable the ChIP data should be extended to other amplicons both within and outside of the 35S transcribed region to show that the reduction in promoter signal is not due to chromatin accessibility changes after nutrient depletion, etc.) As is, the data provided suggest that the reduction in transcription on nutrient withdrawal is not due either to dissociation of the PolI-Rrn3 complex or the formation of inactive PolI dimers.*

We agree with this reviewer that previous figures did not illustrate well the spread of recruitment typically observed in PICT experiments. As mentioned, we have modified the figures to include a zoomed inset around the anchors of a representative cell. In this inset we have color-overlaid segmented spots in the GFP channel, the RFP channel and the intersection of both channels (pixels where we detected co-localization between RFP and GFP). As we have shown for cells expressing A190-GFP, the GFP signal around the anchor in cells expressing Rrn3-GFP is similar in both growing cells and starved cells (Figure 3—figure supplement 1). Rrn3-GFP signal is segmented in the same way as A190-GFP (explained in detail in the Materials and methods section and illustrated in Figure 1—figure supplement 2). The inset with color overlay shows that segmentation of anchors is perfect, and that all apparent GFP spots are detected with some false detections that do not affect significantly the recruitment score since they only marginally occur over an anchor.

Regarding disparity between Pol I–Rrn3 levels and Pol I and Rrn3 promoter association in starved cells, we agree that reduction of Pol I–Rrn3 levels are not sufficient to fully account for inactivation of rDNA transcription. However, we now present a more surgical mutation of the A43 C-terminal tail (new A43ΔCt mutant) where Pol I dimerization is impaired (Figure 1) but the levels of Pol I and Rrn3 promoter association are higher than for the wild-type in starving conditions. This suggests that Pol I homodimerization plays an essential role in inactivation of rDNA transcription. In conclusion, we have tuned-down our claims and now state that both Pol I homodimers and Pol I–Rrn3 complexes modulate rDNA transcription.

To fulfill the reviewer’s requirement, we have performed ChIP experiments at two different regions within the rDNA gene, i.e. 18S and 25S (Figure 3). Our results show that A190 associates with the chromatin all along the transcribed region and the rDNA promoter, while Rrn3 levels are only high at the promoter region for the wild-type strain (Figure 3). However, our ChIP analysis of mutant strains indicates that changes in chromatin accessibility after nutrient depletion do not explain the reduction in Pol I and Rrn3 promoter association. For instance, in the Rrn3-A43 fusion strain, A190 can be localized associated within the 35S rDNA gene under starving conditions at significantly higher levels than in wild-type cells (Figure 3—figure supplement 2). A similar behavior can be observed for the new A43ΔCt mutant (Figure 3). Therefore, while a profound study of chromatin accessibility in different nutrient conditions is very interesting, we feel that a deeper analysis of this event goes far beyond the scope of the current studies and our data do not support this hypothesis.

*The time course data in Figure 4 is potentially interesting, but again is based on a techniques and quantitation regime that are wholly inadequate. Even if we accept that the data do in fact reflect in vivo changes, we have no idea what the relative signals mean in terms of the fraction of PolI in the different complexes. The PolI dimer signal might be rising on nutrient withdrawal and falling on refeeding, but we have no idea how much of PolI is undergoing this change. This is particularly clear from the refeeding data, where dimers appear to be rapidly eliminated, but PolI-Rrn3 complexes hardly increase over the same time period.*

We agree with the reviewer that PICT does not tell about absolute levels of protein complexes but it does reflect relative changes between different time points. As mentioned above, we have now included time-course ChIP experiments showing that, upon refeeding, Pol I is rapidly associated to promoters to activate transcription. Thus, in spite of an increment in the assembly of Pol I–Rrn3 complexes, these will also rapidly disassemble as a result of rRNA synthesis. Therefore, Pol I–Rrn3 levels as seen by PICT are not expected to rise until Pol I–Rrn3 assembly is faster than Pol I incorporation to rDNA, which we observe only at the later stages of recovery from starvation. The absolute quantification of Pol I dimer copies goes beyond the scope of this study. Nonetheless, our novel A43ΔCt mutant shows that Pol I homodimerization is responsible for about 60% of Pol I inhibition in cells starved for 2 hours, highlighting the relevance of this regulatory mechanism.

*In short, the questions are interesting, the technique fascinating, but the data do not support the claims and interpretations.*

*Reviewer #3:*

*[…] Specific comments:*

*For the PICT results, it's sometimes not clear how the individual cell panels illustrate the numbers shown in the bar graphs. For example, in Figure 1, why is the δ 43 heterozygote nearly WT on the bar graph, while the homozygote is zero? The pictures make it look like it should be the other way around. Please clarify exactly what the bar graphs are showing: simple overlap of peaks in two dimensions, or some 3D measurement? Perhaps more pictures would help.*

We have now presented more clear images of cells so that the quantification can be more easily examined. Please refer to responses to reviewer 2 for a thorough explanation on how PICT images have been quantified. The methodology followed to analyze and quantify these assays had not been explained accurately. We have extended the description in the Materials and methods section. We have also modified all the figures to clarify the PICT experiments. Since we had article space limitation and since the spots of the recruited prey-GFP at the anchoring platforms are small, we could not show many cells for each treatment. Instead we selected cells that were representative of each sample. We have adjusted the contrast to facilitate the identification of the recruited prey-GFP. Finally, we have updated the image to include a zoomed inset of the area around the anchoring platforms and also used a color overlay that labels pixels according to segmented spots in both channels.

*In the last paragraph of the subsection “Dynamics in the assembly of Pol I complexes in response to nutrient availability”, the paper states that in the second stage after addition of nutrients, the pol I homodimers disappear, but with no increase in the Rrn3/pol I complex. Where do those polymerase molecules go, if not into the Rrn3 complex?*

As explained for reviewer 2, we interpret that these molecules are engaged in active transcription, so they should be loaded along the rDNA gene. This would explain that there is no apparent change in the overall levels of Pol I–Rrn3 in the middle stage of recovery from starvation.

*I like the idea in final paragraph of the Discussion, which draws parallels between the Rrn3-pol I and Mediator – pol II interactions. However, use of the word "holoenzyme" in the pol II system has been very messy and I would avoid it. To my mind, a holoenzyme should have all the activities needed for promoter recognition and transcription initiation. There were some early papers proposing a pre-assembled holoenzyme of pol II with all the general transcription factors, but those models haven't held up. Later papers started misusing holoenzyme to mean the Mediator – pol II complex. For similar reasons, I don't think Rrn3-pol I would qualify as a holoenzyme. I would avoid the word altogether so as not to distract from the concept that factor interactions with the stalk and surrounding regions might help bring RNA polymerases to the promoter.*

We have now re-written this section to remove the Pol I holoemzyme concept, while keeping the comparison with the Pol II and bacterial systems. The revised manuscript includes a revised paragraph at the end of the Discussion.

[Editors' note: further revisions were requested prior to acceptance, as described below.]

*[…] However, there are two major issues with their description of the results.*

*First, their description of the new coIP experiment shown in Figure 1—figure supplement 4 is very confusing and incomplete. The only results that are straightforward are shown in lanes 1-2 and 5-6, which provide valuable evidence for Pol I homodimers in the soluble fractions obtained from cross-linked cells when the cells have been starved. In their "Responses to reviewers", they provide a somewhat more complete explanation of the last 3 lanes pertaining to the insoluble cell fractions, but even this doesn't make complete sense, and it was left out of the manuscript/legends completely. A coherent explanation of all of the lanes in Figure 1—figure supplement 4 is required.*

We thank the Reviewing editor for this useful recommendation. In addition to a more detailed description of the coIP experiments, labelling of lanes 8 and 9 in Figure 1—figure supplement 4 had been interchanged and is now amended. The revised version of the manuscript now reads:

“In addition, we performed co-immunoprecipitation experiments after crosslinking, using a diploid strain where one A190 allele was tagged with TAP (A190-TAP) and the second with MYC (A190-MYC). […] The absence of histone H3 in the soluble fraction indicates that there is no contamination from the chromatin insoluble fraction (Figure 1—figure supplement 4).”

*Second, in the Discussion: the statement that "initial transcriptional inactivation is mainly driven by Pol I:Rrn3 disassembly." does not seem justified. The experiments in Figure 4 clearly show that the bulk of the dissociation of Pol I: Rrn3 from the promoter occurs within the first 20 min at a timepoint where Pol I:Rrn3 complexes are still very abundant and almost no Pol I homodimers have formed. It seems unavoidable that there is another regulatory mechanism apart from dissociation of Pol I:Rrn3 complexes and Pol I homodimerization responsible for the rapid and nearly complete loss of promoter-bound Pol I on starvation. From their statements at several places in the text, it appears that the authors actually agree with this view, but they don't state it directly in the Discussion and it is not depicted in the final model, which could be misleading to the field. The authors are urged to acknowledge more explicating in the Discussion and in their final model that there remains an important gap in knowledge about how the rapid dissociation of the Pol I:Rrn3 complex from the promoter is achieved on starvation.*

We agree with the view of the Reviewing editor and have now incorporated this concept in the Discussion section, as well as in the new Figure 8 and accompanying legend. The revised manuscript now reads:

“When nutrients are depleted, Pol I–Rrn3 levels and promoter association drop exponentially whereas Pol I only homodimerizes subsequently. […] At a later stage, Pol I homodimerization remains a major factor limiting transcription.”

“Figure 8. Model for the influence of nutrient availability on the assembly of Pol I complexes. […] Transparency indicates mobility of the stalk in monomeric Pol I.”